# Direct Molecular Conformation Generation*

**Jinhua Zhu**[1]                                        *teslazhu@mail.ustc.edu.cn*
**Yingce Xia**[2]                                        *yingce.xia@microsoft.com*
**Chang Liu**[2]                                         *changliu@microsoft.com*
**Lijun Wu**[2]                                          *lijuwu@microsoft.com*
**Shufang Xie**[3]                                       *shufangxie@ruc.edu.cn*
**Yusong Wang**[4]                            *wangyusong2000@stu.xjtu.edu.cn*
**Tong Wang**[2]                                         *watong@microsoft.com*
**Tao Qin**[2]                                           *taoqin@microsoft.com*
**Wengang Zhou**[1]                                      *zhwg@ustc.edu.cn*
**Houqiang Li**[1]                                       *lihq@ustc.edu.cn*
**Haiguang Liu**[2]                                      *liuhaiguang@microsoft.com*
**Tie-Yan Liu**[2]                                       *tyliu@microsoft.com*
[1] *University of Science and Technology of China*   [2] *Microsoft Research AI4Science*
[3] *Renmin University of China*   [4] *Xi'an Jiaotong University*

**Reviewed on OpenReview:** *https://openreview.net/forum?id=lCPOHiztuw*

## Abstract

Molecular conformation generation aims to generate three-dimensional coordinates of all the atoms in a molecule and is an important task in bioinformatics and pharmacology. Previous methods usually first predict the interatomic distances, the gradients of interatomic distances or the local structures (e.g., torsion angles) of a molecule, and then reconstruct its 3D conformation. How to directly generate the conformation without the above intermediate values is not fully explored. In this work, we propose a method that directly predicts the coordinates of atoms: (1) the loss function is invariant to roto-translation of coordinates and permutation of symmetric atoms; (2) the newly proposed model adaptively aggregates the bond and atom information and iteratively refines the coordinates of the generated conformation. Our method achieves the best results on GEOM-QM9 and GEOM-Drugs datasets. Further analysis shows that our generated conformations have closer properties (e.g., HOMO-LUMO gap) with the groundtruth conformations. In addition, our method improves molecular docking by providing better initial conformations. All the results demonstrate the effectiveness of our method and the great potential of the direct approach. The code is released at `https://github.com/DirectMolecularConfGen/DMCG`.

## 1   Introduction

Molecular conformation generation aims to generate 3D atomic coordinates of a molecule, which then can be used in molecular property prediction (Axelrod & Gomez-Bombarelli, 2021), docking (Roy et al., 2015), structure-based virtual screening (Kontoyianni, 2017), etc. While molecular conformation is experimentally obtainable, such as via X-ray crystallography, it is prohibitively costly for industry-scale tasks (Mansimov et al., 2019). *Ab initio* methods, e.g., based on density functional theory (DFT) (Parr, 1980; Baseden & Tye, 2014), can accurately predict molecular structures, but take several hours per small molecule (Hu et al., 2021). To handle large molecules, people turn to leverage classical force fields, like UFF (Rappe et al., 1992) or MMFF (Halgren, 1996) and its extension (Cleves & Jain, 2017), to optimize conformations, which is efficient but at the cost of low accuracy (Kanal et al., 2018).

---

*This work was done when Jinhua Zhu and Yusong Wang were interns at Microsoft Research AI4Science. Correspondence to Yingce Xia and Chang Liu.

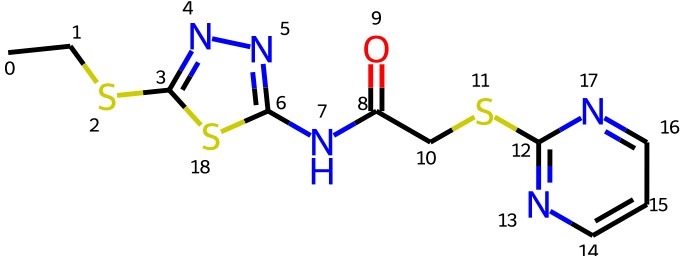

Figure 1: An example of symmetric substructure of a molecule.

Recently, machine learning methods have attracted much attention for conformation generation due to their accuracy and efficiency. Most of previous methods first predict some intermediate values, like interatomic distances (Simm & Hernández-Lobato, 2020; Shi et al., 2020; Xu et al., 2021a;b), the gradients w.r.t. interatomic distances (Shi et al., 2021; Luo et al., 2021b), or the torsion angles (Ganea et al., 2021), and then reconstruct the conformation based on them. While those methods improve molecular conformation generation, the intermediate values they used should satisfy additional hard constraints, which are unfortunately violated in many cases. For example, GraphDG (Simm & Hernández-Lobato, 2020) predicts the interatomic distances and then reconstructs the conformation based on them. The real distances (e.g., considering three atoms) should satisfy the triangle inequality, but the distances predicted by GraphDG violate the inequality out of 8.65% cases according to our study. For another example, ConfGF (Shi et al., 2021) predicts the gradient of interatomic distances, and the rank of a squared distance matrix is at most five. Such constraint makes gradients ill-defined, because other distances cannot all be held constant while taking an infinitesimal change to a specific distance $d_{ij}$ (see Appendix C for more details). Directly generating the coordinates without those intermediate values is a more straightforward strategy but is not fully explored. AlphaFold 2 (Jumper et al., 2021) is such a kind of direct approach and has achieved remarkable performances on protein structure prediction. The success of AlphaFold 2 inspires us to explore the method of directly generating coordinates for molecular conformation.

A challenge of this approach is to maintain roto-translation invariance and permutation invariance. Specifically, (1) rotating and translating the coordinates of all atoms as a group do not change the conformation of a molecule, which should be taken into consideration for the direct approach; (2) Permutation invariance should be considered for symmetry-related atoms. For example, as shown in Figure 1, due to the symmetry of the pyrimidine part along the C-S bond (atom 11 and 12), atoms $13, 14$ and atoms $17, 16$ are equivalent. Therefore, swapping the coordinates of 13 with 17 and 14 with 16 yields the same conformation. According to our statistics on a subset of 40K molecules from GEOM-Drugs (Axelrod & Gomez-Bombarelli, 2021), on average, each molecule has 5.9 atom mappings which could result in the same conformation (more specifically, the average number of $|\mathcal{S}|$ in Eqn.(1) is 5.9). The number is non-negligible for loss function design.

To maintain roto-translation and permutation invariance, in our method, we design a loss function as the minimal distance between two sets of coordinates after any roto-translation and permutation of symmatric atoms. Based on the new loss function, we design a model that iteratively refines atom coordinates. The model stacks multiple blocks, and each block outputs a conformation which is then refined by the following block. A block consists of several modules that encode the previous conformation as well as the representations of bonds, atoms and global information of molecules. At the end of each block, we add a normalization layer that centers the coordinates at the origin. Since a molecule may have multiple conformations, inspired by variational auto-encoder (VAE), we introduce a random variable $z$ and a regularization term on $z$, which allows diverse generation.

We conduct experiments on four benchmarks: GEOM-QM9 and GEOM-Drugs with the small-scale setting (Shi et al., 2021) and large-scale setting (Axelrod & Gomez-Bombarelli, 2021). The small-scale GEOM-QM9 and GEOM-Drugs have 200K molecule-conformation pairs for training, and the large-scale GEOM-QM9 and GEOM-Drugs have 1.37M and 2.0M training pairs. Our method achieves state-of-the-art results on all of them, demonstrating the effectiveness of our method. Specifically, on small-scale GEOM-QM9, our method improves the recall-based mean coverage score and mean matching score by 4.7% and 0.3%. On

small-scale GEOM-Drugs, the improvements are 7.4% and 16.3%. On the large-scale settings, the improvements are more significant: 7.3% and 47.1% for GEOM-QM9, and 25.3% and 36.0% for GEOM-Drugs. To further verify the generation quality, we use Psi4 (Smith et al., 2020) to calculate the properties of generated conformations and groundtruth conformations (e.g., HOMO-LUMO gap). Our conformations have closer properties to the groundtruth compared with other methods. We also find that our generated conformations can help improve molecular docking by providing better initial conformations.

To summary, (1) we design a dedicated loss function, that can maintains both permutation invariance on symmetric atoms and roto-translation invariance on conformations; (2) we design a new model that iteratively refines the conformation. (3) our method, named Direct Molecular Conformation Generation (DMCG), outperforms strong baselines and achieves state-of-the-art results on all benchmarks we tested.

**Problem Definition**: Let $G = (V, E)$ denote a molecular graph, where $V$ and $E$ are collections of atoms and bonds, respectively. Specifically, $V = \{v_1, v_2, \cdots, v_{|V|}\}$ with the $i$-th atom $v_i$. Let $e_{ij}$ denote the bond between atom $v_i$ and $v_j$. For ease of reference, we simply use $i \in V$ and $(i, j) \in E$ to denote the $i$-th atom in $V$ and the bond $e_{ij}$ in $E$. Let $N(i)$ denote the neighbors of atom $i$, i.e., $N(i) = \{j \mid (i, j) \in E\}$. We use $R$ to represent the conformation of $G$, where $R \in \mathbb{R}^{|V| \times 3}$. The $i$-th row of $R$ (denoted as $R_i$) is the coordinate of atom $v_i$. Given a graph $G = (V, E)$, our task is to learn a mapping, that can output the coordinates $R$ of all atoms in $V$, i.e., $R \in \mathbb{R}^{|V| \times 3}$.

## 2 Framework

In this section, we first introduce the loss function. After that, we present the overall training and inference workflow of our method. Finally, we introduce our proposed model.

### 2.1 Loss function

**Roto-translational and permutation invariance of conformations**: Let $R \in \mathbb{R}^{|V| \times 3}$ and $\hat{R} \in \mathbb{R}^{|V| \times 3}$ denote the groundtruth conformation and the generated conformation. The roto-translation and permutation invariant loss is defined as follows:

$$\ell_{\mathrm{RTP}}(R, \hat{R}) = \min_{\rho; \, \sigma \in \mathcal{S}} \|R - \rho(\sigma(\hat{R}))\|_F^2. \tag{1}$$

In Eqn.(1), (i) $\rho$ denotes a roto-translational operation, which means to rotate and translate a conformation rigidly; (ii) $\mathcal{S}$ denotes the collection of the permutation operations on symmetric atoms. For example, in Figure 1, $\mathcal{S}$ contains two elements $\sigma_1$ and $\sigma_2$, where $\sigma_1$ is an identical mapping, i.e. $\sigma_1(i) = i$ for any $i \in \{1, 2, \cdots, 18\}$, and $\sigma_2$ is the mapping on symmetric atoms of the pyrimidine: $\sigma_2(13) = 17, \sigma_2(17) = 13, \sigma_2(14) = 16, \sigma_2(16) = 14$ and $\sigma_2(i) = i$ for the remaining atom $i$'s. (iii) $\|A\|_F^2$ is defined as $\sum_{i,j} |A_{i,j}|^2$. In all, Eqn.(1) defines a loss between $R$ and $\hat{R}$ as the minimal achievable distance under any roto-translation operation and any permutation operation of symmetric atoms, hence is invariant to these operations. Eqn.(1) can be solved via quaternions (Karney, 2007; Hamilton, 1840) and graph isomorphism (Meli & Biggin, 2020).

To solve Eqn. (1), the optimization can be decomposed into two sub problems: (S1) $\ell_{\mathrm{RT}} = \min_\rho \|\rho(\hat{R}) - R\|_F^2$; (S2) $\ell_{\mathrm{P}} = \min_{\sigma \in \mathcal{S}} \|\sigma(\hat{R}) - R\|_F^2$.

Karney (2007) propose to use quaternions (Hamilton, 1840) to solve (S1). A quaternion $q$ is an extension of complex numbers, $q = q_0\mathbf{u} + q_1\mathbf{i} + q_2\mathbf{j} + q_3\mathbf{k}$, where $q_0, q_1, q_2, q_3$ are real scalars and $\mathbf{u}, \mathbf{i}, \mathbf{j}, \mathbf{k}$ are orientation vectors. With quaternions, any rotation operation is specified by a $3 \times 3$ matrix, where each element in the matrix is the summation/multiplication of $q_0$ to $q_3$. The solution to (S1) is the minimal eigenvalue of a $4 \times 4$ matrix obtained by algebraic operations on $R$ and $\hat{R}$. To stabilize training, we stop gradient back-propagation through $\rho$ (see Appendix B.3 for the ablation study).

To solve (S2), we need to find all elements in $\mathcal{S}$, and then enumerate them to get the minimal value. $\mathcal{S}$ can be mathematically described as follows: (1) $\forall i \in V$, atom $i$ and atom $\sigma(i)$ have the same label, which is defined as the union of the atom type itself and also the types of all the bonds connected to it[1]. (2) There

---

[1]For example, in Figure 1, the label of atom 11 is "S-2Single", and the label of atom 17 is "N-2Aromatic".

exists a bond between atoms $i$ and $j$ if and only if there exists a bond between atoms $\sigma(i)$ and $\sigma(j)$ in the same molecular graph. Therefore, we convert finding $\mathcal{S}$ into a graph isomorphism problem on molecular graphs. Inspired by Meli & Biggin (2020), we use the `graph_tool` toolkit[2] to find all permutations in $\mathcal{S}$. By combining the above two strategies, we are able to solve Eqn.(1). We provide several examples in the online supplementary material to show how our method works.

Hopcroft & Wong (1974) proposed an algorithm whose complexity of testing planar graphs for isomorphism is $O(|E|)$, where $|E|$ is the number of edges in a graph. A planar graph can be regarded as a type of graph that no edges cross each other (see Wiki for a quick introduction). For the widely used GEOM-QM9 and GEOM-Drugs datasets (Shi et al., 2021; Xu et al., 2022) of conformation generation, all the molecules are planar graphs. We also randomly sample 30M compounds from PubChem, and only 4.5k of them are not planar graphs (0.015%). This shows that although our method needs to test graph isomorphism, the time complexity could still be controlled. In addition, the $|\mathcal{S}|$'s of 99.8% molecules in GEOM-Drugs are smaller than 100 and efficient to enumerate them all in GPU. A limitation is that, when stepping from small molecules to proteins with a long chain, $|\mathcal{S}|$ will significantly increase, resulting in large computation cost of obtaining $\ell_{\text{RTP}}$. We will improve it in the future.

**One-to-many mapping of conformations**: A molecule might correspond to multiple conformations. Thus, we introduce a random variable $z$ to our model for diverse conformation generation. Given a molecular graph $G$, different $z$ could result in different conformations (denoted as $\hat{R}(z, G)$). Inspired by the variational auto-encoder (VAE) (Kingma & Welling, 2014; Rezende et al., 2014; Sohn et al., 2015), we introduce a (conditional) inference model $q(z|R, G)$ to describe the posterior distribution of $z$, reform the reconstruction loss in a probabilistic style $\mathbb{E}_{q(z|R,G)}\left[\ell_{\text{RTP}}(R, \hat{R}(z, G))\right]$, and append a regularization term in the form of the Kullback-Leibler (KL) divergence w.r.t. a prior distribution $p(z)$, i.e. $D_{\text{KL}}(q(z|R,G)\|p(z))$. In this way, the aggregated (i.e. averaged/marginalized) posterior $\int q(z|R,G)p_{\text{data}}(R)\,\mathrm{d}R$ is driven towards the prior $p(z)$, which in turn allows generating a new conformation from $p_{\text{data}}(R)$ by passing through the decoder with a $p(z)$ sample. It is easy to draw a random variable $z$ from $p(z)$ and encourages diversity.

By properly choosing $q(z|R, G)$, the loss is tractable to optimize. We specify $q(z|R,G) := \mathcal{N}(z|\mu_{R,G}, \Sigma_{R,G})$ as a multivariate Gaussian with a diagonal covariance matrix, where the $\mu_{R,G}$ and $\Sigma_{R,G}$ are outputs from an encoder. It enables tractable loss optimization via reparameterization (Kingma & Welling, 2014): $z \sim q(z|R,G)$ is equivalent to $z^{(i)} = \mu_{R,G}^{(i)} + \sqrt{\Sigma_{R,G}^{(i,i)}}\epsilon, \forall i$, where $\epsilon \sim \mathcal{N}(0, 1)$, $z^{(i)}$ and $\mu_{R,G}^{(i)}$ are the $i$-th element of $z$ and $\mu_{R,G}$, and $\Sigma_{R,G}^{(i,i)}$ denotes the $i$-th diagonal element. The KL divergence loss is specialized as $D_{\text{KL}}(\mathcal{N}(\mu_{R,G}, \Sigma_{R,G})\|\mathcal{N}(0, \boldsymbol{I}))$, which has closed form solution.

Overall, the overall training objective function is defined as follows:

$$\min \quad \mathbb{E}_{\epsilon \sim \mathcal{N}(0,\boldsymbol{I})}\ell_{\text{RTP}}(R, \hat{R}(z, G)) + \beta D_{\text{KL}}(\mathcal{N}(\mu_{R,G}, \Sigma_{R,G})\|\mathcal{N}(0, \boldsymbol{I})), \tag{2}$$

where $\beta > 0$ is a hyperparameter. The minimization in Eqn.(2) is taken over all the network parameters (including the conformation generator and auxiliary model $q$).

## 2.2 Training and inference flow

Now we show the training and inference workflow. The training process involves three modules, $\varphi_{\text{2D}}$, $\varphi_{\text{3D}}$ and $\varphi_{\text{dec}}$. The workflow is illustrated in Figure 2(a). Specifically,

(1) The encoder $\varphi_{\text{2D}}$ takes the molecular graph $G$ as its input, and outputs several representations: $H_V^{(0)} \in \mathbb{R}^{|V| \times d}$ for all atoms, $H_E^{(0)} \in \mathbb{R}^{|E| \times d}$ for all bonds, a global graph feature $U^{(0)} \in \mathbb{R}^d$, and initial conformation $\hat{R}^{(0)} \in \mathbb{R}^{|V| \times 3}$. Note $d$ is the dimension of the representations. Formally, $(H_V^{(0)}, H_E^{(0)}, U^{(0)}, \hat{R}^{(0)}) = \varphi_{\text{2D}}(G)$.

(2) The encoder $\varphi_{\text{3D}}$ extracts features of the conformation $R$ for constructing the conditional inference module $q(z|R, G)$. According to the above specification, $\varphi_{\text{3D}}$ only needs to output the mean and covariance of the Gaussian, or formally, $(\mu_{R,G}, \Sigma_{R,G}) = \varphi_{\text{3D}}(R, G)$.

---

[2]https://graph-tool.skewed.de

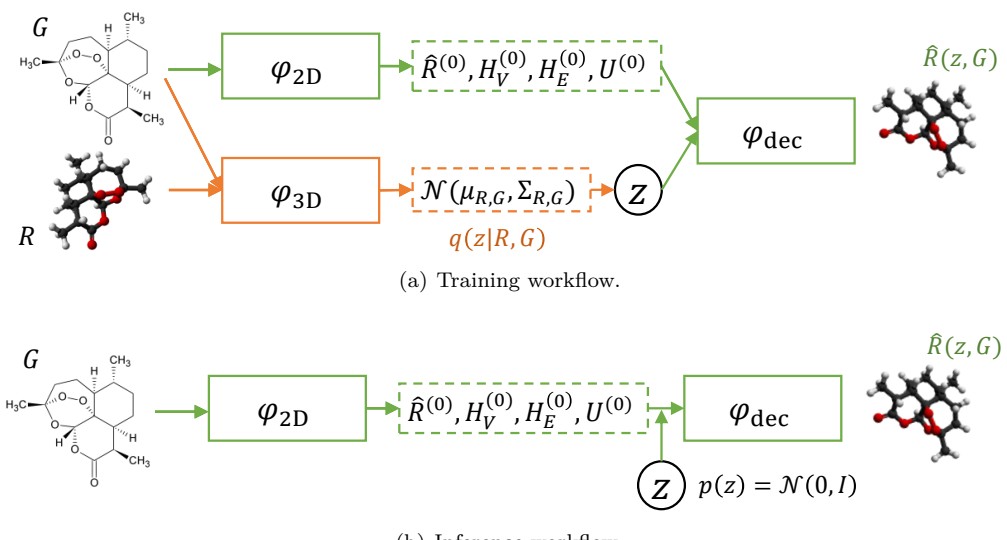

(a) Training workflow.

(b) Inference workflow.

Figure 2: The workflow of our method. Green and orange lines represent how to obtain $\hat{R}(z, G)$ and $q(z|R, G)$ respectively. Solid lines and dashed lines represent the model components and outputs respectively.

(3) We randomly sample a variable $z$ from the Gaussian distribution $\mathcal{N}(\mu_{R,G}, \Sigma_{R,G})$, and then feed $H_V^{(0)}$, $H_E^{(0)}$, $U^{(0)}$, $\hat{R}^{(0)}$, $z$ into the decoder $\varphi_{\text{dec}}$ to obtain the conformation $\hat{R}(z, G)$. That is, $\hat{R}(z, G) = \varphi_{\text{dec}}(\varphi_{\text{2d}}(G), z) = \varphi_{\text{dec}}(H_V^{(0)}, H_E^{(0)}, U^{(0)}, \hat{R}^{(0)}, z)$. Note that sampling $z \sim \mathcal{N}(\mu_{R,G}, \Sigma_{R,G})$ is equivalent to sampling $\epsilon \sim \mathcal{N}(0, \boldsymbol{I})$ and then setting $z^{(i)} = \mu_{R,G}^{(i)} + \sqrt{\Sigma_{R,G}^{(i,i)}}\epsilon$.

(4) After obtaining $\hat{R}(z, G)$ and $\mathcal{N}(\mu_{R,G}, \Sigma_{R,G})$, we optimize Eqn.(2) for training. Recall that $\hat{R}(z, G)$ is related to $\varphi_{\text{2D}}, \varphi_{\text{3D}}, \varphi_{\text{dec}}$, and $\mu_{R,G}, \Sigma_{R,G}$ are related to $\varphi_{\text{3D}}$.

The inference workflow is shown in Figure 2(b), where the well-trained $\varphi_{\text{2D}}$ and $\varphi_{\text{dec}}$ are leveraged: (1) Given a molecular graph $G$, we use $\varphi_{\text{2D}}$ to encode $G$ and obtain $\hat{R}^{(0)}$, $H_V^{(0)}$, $H_E^{(0)}$, $U^{(0)}$; (2) we sample a random variable $z$ from Gaussian $\mathcal{N}(0, \boldsymbol{I})$; (3) we feed $\hat{R}^{(0)}$, $H_V^{(0)}$, $H_E^{(0)}$, $U^{(0)}$, $z$ into $\varphi_{\text{dec}}$ and obtain the eventual conformation $\hat{R}(z, G)$. Note that $\varphi_{\text{3D}}$ is not used in inference phase.

## 2.3 Model architecture

The encoders $\varphi_{\text{2D}}, \varphi_{\text{3D}}$ and the decoder $\varphi_{\text{dec}}$ share the same architecture. They all stack $L$ identical blocks. We take the decoder $\varphi_{\text{dec}}$ as an example to introduce its $l$-th block, and leave the details of $\varphi_{\text{2D}}$ and $\varphi_{\text{3D}}$ to Appendix A.1.

Figure 3 shows the architecture of the $l$-th block of $\varphi_{\text{dec}}$. Roughly speaking, this block takes the outputs from its preceding block (including the conformation $\hat{R}^{(l-1)}$, atom representations $H_V^{(l-1)}$, edge representations $H_E^{(l-1)}$ and the global representation $U^{(l-1)}$ of the whole molecule) and outputs refined conformation and representations of atoms, bonds, the whole graph. The process is repeated until the eventual output $\hat{R}^{(L)}$ is obtained. For the input of the first block (i.e., $l = 1$), the $H_V^{(0)}$, $H_E^{(0)}$, $U^{(0)}$ and $\hat{R}^{(0)}$ are the outputs of $\varphi_{\text{2D}}$.

We use a variant of the GN block (Battaglia et al., 2018; Addanki et al., 2021) as the backbone of our model due to its superior performance in molecular modeling. In each block, we first update bond representations, then atom representations, and finally the global molecule representation and the conformation. For ease of reference, let $h_i^{(l)}$ denote the representation of atom $i$ output by the $l$-th block, and $h_{ij}^{(l)}$ the representation of the bond between atom $i$ and $j$. Also, let MLP denote a feed-forward network.

Mathematically, the $l$-th block takes following operations:

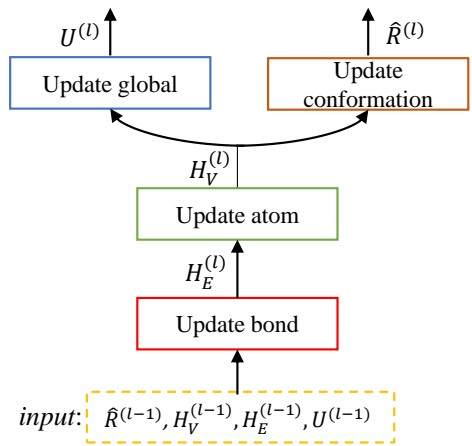

Figure 3: Network architecture of the $l$-th block.

(1) *Update bond representations*: We first incorporate the coordinate information into the representations by

$$\bar{h}_i^{(l)} = h_i^{(l-1)} + \texttt{MLP}(\hat{R}_i^{(l-1)}) + z, \ \forall i \in V, \tag{3}$$
$$\bar{h}_{ij}^{(l)} = h_{ij}^{(l-1)} + \texttt{MLP}(\|\hat{R}_i^{(l-1)} - \hat{R}_j^{(l-1)}\|), \ \forall (i,j) \in E,$$

where $z \sim \mathcal{N}(\mu_{R,G}, \Sigma_{R,G})$. After that, the bond representations are updated as follows: $\forall (i,j) \in E$,

$$h_{ij}^{(l)} = h_{ij}^{(l-1)} + \texttt{MLP}(\bar{h}_i^{(l-1)}, \bar{h}_j^{(l-1)}, \bar{h}_{ij}^{(l-1)}, U^{(l-1)}). \tag{4}$$

(2) *Update atom representations*: for any atom $i \in V$,

$$\tilde{h}_i^{(l)} = \sum_{j \in N(i)} \alpha_j W_v \texttt{concat}(\bar{h}_{ij}^{(l)}, \bar{h}_j^{(l-1)}) \text{ where } \alpha_j \propto \exp(\boldsymbol{a}^\top \zeta(W_q \bar{h}_i^{(l-1)} + W_k \texttt{concat}(\bar{h}_j^{(l-1)}, \bar{h}_{ij}^l)));$$
$$h_i^{(l)} = h_i^{(l-1)} + \texttt{MLP}\left(\bar{h}_i^{(l-1)}, \tilde{h}_i^{(l)}, U^{(l-1)}\right). \tag{5}$$

In Eqn.(5), $\boldsymbol{a}$, $W_q$, $W_v$ and $W_k$ are the parameters to be learned, $\texttt{concat}(\cdot, \cdot)$ is the concatenation of two vectors and $\zeta$ is the leaky ReLU activation. For atom $v_i$, we first use GATv2 (Brody et al., 2021) to aggregate the representations from its connected bonds to obtain $\tilde{h}_i$, and then update $v_i$ based on $\tilde{h}_i^{(l)}$, $\bar{h}_i^{(l-1)}$ and $U^{(l-1)}$.

(4) *Update global molecule representation*:

$$U^{(l)} = U^{(l-1)} + \texttt{MLP}\left(\frac{1}{|V|}\sum_{i=1}^{|V|} h_i^{(l)}, \frac{1}{|E|}\sum_{i,j} h_{ij}^{(l)}, U^{(l-1)}\right). \tag{6}$$

(5) *Update the conformation*: $\forall i \in V$,

$$\bar{R}_i^{(l)} = \texttt{MLP}(h_i^{(l)}), \quad m^{(l)} = \frac{1}{|V|}\sum_{j=1}^{|V|} \bar{R}_j^{(l)}, \quad \hat{R}_i^{(l)} = \bar{R}_i^{(l)} - m^{(l)} + \hat{R}_i^{(l-1)}. \tag{7}$$

An important step in Eqn.(7) is that, after making initial prediction $\bar{R}_i^{(l)}$, we calculate its center and normalize their coordinates by moving the center to the origin. This normalization ensures that the coordinates generated by each block are in reasonable numeric ranges.

We use $\hat{R}^{(L)}$ output by the last block in $\varphi_{\text{dec}}$ as the final prediction of the conformation.

## 3 Discussions with related work

CVGAE (Mansimov et al., 2019) is an early attempt to directly generating conformation. Unfortunately, its performance is not as good as distance-based methods developed afterwards, like (Shi et al., 2020) and

(Simm & Hernández-Lobato, 2020). Our method, pursuing the same spirit, makes several finer designs: (1) We design a dedicated training objective that takes the invariance of both roto-translation and permutation on symmetric atoms into consideration. (2) We iteratively refine the output of each block, which is effective for conformation generation (see Figure 7 for ablation study). In comparison, CVGAE only outputs the conformation in the last layer. (3) Our model integrates several advanced and more effective modules, including GATv2 (Brody et al., 2021) and GN block (Battaglia et al., 2018), while CVGAE mainly leverages GRU (Bahdanau et al., 2015) and its variants on graphs, which are outperformed by the modules used in our model. GeoDiff (Xu et al., 2022) is a concurrent work, which uses a diffusion-based method for conformation generation and also directly predicts the coordinates without using intermediate distances. Compared with our method, GeoDiff does not consider the permutation invariance of symmetric atoms and is not as efficient as our method due to its sequential sampling.

ConfGF (Shi et al., 2021) and DGSM (Luo et al., 2021b) are two recent works that can also directly output the coordinates. They both model the gradient of log-density w.r.t interatomic distances, and then generate coordinates by running Langevin dynamics using the gradients. The gradient model is learned via score-matching. ConfGF considers the distances of 1-hop, 2-hop and 3-hop neighbors, and DGSM also considers distances of two randomly sampled nodes to model non-bonded distances. In comparison, we completely get rid of modeling distances. More importantly, the permutation invariance of symmetric atoms are not considered in those works. Ganea et al. (2021) propose another method for conformation generation: they first build the local structure (LS) by predicting the coordinates of non-terminal atoms, and then refine the LS by the predicted distances and dihedral angles. In comparison, our method does not require refinement based on the predicted distances and angles. Furthermore, although Ganea et al. (2021) use a permutation invariant loss, they only consider the terminal atoms. According to our statistics on a subset of $40K$ molecules from GEOM-Drugs, besides terminal atoms, on average, a molecule has 4.9 non-terminal symmetric atoms, accounting for 10.8% of all atoms. We consider all symmetric atoms.

Our method models the roto-translation and permutation invariance through the loss function, while previous works model the molecules using equivariant networks (Hoogeboom et al., 2022; Xu et al., 2022). More specifically, these works use the diffusion model for conformation generation. Rotational invariance of the conformation distribution is implemented using an invariant latent prior and an equivariant model structure (reverse diffusion process) to map from the latent space to the conformation space. This effectively makes an invariant loss in the latent space. Hoogeboom et al. (2022) also generate the composition of a molecule, by leveraging continuous representation of ordinal/categorical variables. In comparison, our method removes the constraints on equivariant neural networks by introducing equivariance/invariance into loss function, which is different from previous works that rely on specific network designs to ensure equivariance/invariance. A recent work (Du et al., 2022) points that only using radial direction to represent the geometric information (like the models used in Hoogeboom et al. (2022) and Xu et al. (2022)) abandons high-order tensor information, thus bringing direction degeneration problem and is insufficient to express complex geometric qualities. Therefore, in our approach, we can adopt both equivariant network models and more general (non-equivariant) networks, enabling the possibility of using more powerful non-equivariant neural models.

There are some other works on conformation generation, but they target at different problems. G-SchNet (Gebauer et al., 2019; Hoogeboom et al., 2022) takes some properties as input (not 2D graph) and output a conformation with desired properties. Luo et al. (2021a) focus on generating a conformation that can bind with specific binding pocket. We can combine our method with them in the future.

## 4 Experiments

### 4.1 Settings

*Datasets*: Following prior works (Xu et al., 2021a; Shi et al., 2021), we use the GEOM-QM9 and GEOM-Drugs datasets (Axelrod & Gomez-Bombarelli, 2021) for conformation generation. We verify our method on both small-scale setting and large-scale setting. For the small-scale setting, we use the same datasets provided by Shi et al. (2021) for fair comparison with prior works. The training, validation and test sets of

the two datasets consist of 200K, 2.5K and 22408 (for GEOM-QM9)/14324 (for GEOM-Drugs) molecule-conformation pairs respectively. After that, we work on the large-scale setting by sampling larger datasets from the original GEOM to validate the scalability of our method. We use all data in GEOM-QM9 and $2.2M$ molecule-conformation pairs for GEOM-Drugs. The numbers of training, validation and test sets for the larger GEOM-QM9 setting are 1.37M, 165K and 174K, and those for larger GEOM-Drugs are 2M, 100K and 100K.

*Model configuration*: All of $\varphi_{2D}$, $\varphi_{3D}$ and $\varphi_{dec}$ have 6 blocks. The dimension $d$ of the features is 256. Inspired by the feed-forward layer in Transformer (Vaswani et al., 2017), `MLP` also consists of two sub-layers, where the first one maps the input features from dimension 256 to hidden states, followed by Batch Normalization and ReLU activation. Then the hidden states is mapped to 256 again using linear mapping. Considering that our method outputs a conformation $\hat{R}^{(l)}$ at each block $l$, we also require that each $\hat{R}^{(l)}$ should try to be similar to the groundtruth $R$. Therefore, the $\ell_{RTP}$ is Eqn.(2) is implemented as

$$\ell_{RTP}(\hat{R}^{(L)}, R) + \lambda \sum_{l=0}^{L-1} \ell_{RTP}(\hat{R}^{(l)}, R),\qquad(8)$$

where $L$ is the number of blocks in the decoder, $\hat{R}^{(0)}$ is the output from $\varphi_{2D}$, and $\lambda$ is determined according to validation performance. More details are summarized in Appendix A.2.

*Evaluation*: Assuming in the test set, the molecule $x$ has $N_x$ conformations. Following Shi et al. (2020; 2021), for each molecule $x$ in the test set, we generate $2N_x$ conformations. Let $\mathbb{S}_g$ and $\mathbb{S}_r$ denote all generated and groundtruth conformations respectively. We use coverage score (COV) and matching score (MAT) to evaluate the generation quality. To measure the difference between $R$ and $\hat{R}$, we use the `GetBestRMS` in the `RDKit` package and denote the root-mean-square deviation as $\text{RMSD}(R, \hat{R})$. The recall-based coverage and matching scores are defined as follows:

$$\text{COV}(\mathbb{S}_g, \mathbb{S}_r) = \frac{1}{|\mathbb{S}_r|} \left| \left\{ R \in \mathbb{S}_r \mid \text{RMSD}(R, \hat{R}) < \delta, \exists \hat{R} \in \mathbb{S}_g \right\} \right|;$$
$$\text{MAT}(\mathbb{S}_g, \mathbb{S}_r) = \frac{1}{|\mathbb{S}_r|} \sum_{R \in \mathbb{S}_r} \min_{\hat{R} \in \mathbb{S}_g} \text{RMSD}(R, \hat{R}).\qquad(9)$$

A good method should have a high COV score and a low MAT score. Following (Shi et al., 2021; Xu et al., 2022), the $\delta$'s are set as 0.5 and 1.25 for GEOM-QM9 and GEOM-Drugs, respectively. The COV-$\delta$ curves are left in Figure 9 of the appendix. There are also precision-based COV and MAT scores by switching the $\mathbb{S}_r$ and $\mathbb{S}_g$ in Eqn.(9). We leave the precision-based results in Appendix B.1.

*Baselines*: (1) RDKit, which is a widely used toolkit and generates the conformation based on the force fields; (2) CVGAE (Mansimov et al., 2019), which is an early attempt to generate raw coordinates; (3) GraphDG (Simm & Hernández-Lobato, 2020), a representative distance-based method with VAE; (4) CGCF (Xu et al., 2021a), which is another distance-based method leveraging continuous normalizing flow; (5) ConfVAE (Xu et al., 2021b), an end-to-end framework for molecular conformation generation, which still uses the pairwise distances among atoms as intermediate variables; (6) ConfGF (Shi et al., 2021) and DGSM (Luo et al., 2021b), which uses score matching to generate the gradients w.r.t distances and then recover the conformation; (7) GeoDiff (Xu et al., 2022), which uses diffusion model to generate conformations; (8) GeoMol (Ganea et al., 2021), which predicts local atomic 3D structures and torsion angles. Considering Ganea et al. (2021) use a different data split from previous work, we reproduce their method following the more commonly used data split (Xu et al., 2021a; Shi et al., 2021).

## 4.2 Results

The recall-based results are shown in Table 1. For small-scale datasets, we independently train our models with five different random seeds, and report the mean and standard derivations. We have the following observations:

(1) On the four settings in Table 1, our method achieves state-of-the-art results on all of them. The median COV(%) being 100% means that for more than half of the groundtruth conformations, there exist generated

Table 1: Recall-based coverage and matching scores. Bold fonts indicate the best results. The standard derivations of our method (five independent runs) on small-scale datasets are reported.

| Methods | Small-scale QM9 | | | | Small-scale Drugs | | | |
| | COV(%)↑ | | MAT (Å)↓ | | COV(%)↑ | | MAT (Å)↓ | |
| | Mean | Median | Mean | Median | Mean | Median | Mean | Median |
|---|---|---|---|---|---|---|---|---|
| RDKit | 83.26 | 90.78 | 0.3447 | 0.2935 | 60.91 | 65.70 | 1.2026 | 1.1252 |
| CVGAE | 0.09 | 0.00 | 1.6713 | 1.6088 | 0.00 | 0.00 | 3.0702 | 2.9937 |
| GraphDG | 73.33 | 84.21 | 0.4245 | 0.3973 | 8.27 | 0.00 | 1.9722 | 1.9845 |
| CGCF | 78.05 | 82.48 | 0.4219 | 0.3900 | 53.96 | 57.06 | 1.2487 | 1.2247 |
| ConfVAE | 80.42 | 85.31 | 0.4066 | 0.3891 | 53.14 | 53.98 | 1.2392 | 1.2447 |
| GeoMol | 71.26 | 72.00 | 0.3731 | 0.3731 | 67.16 | 71.71 | 1.0875 | 1.0586 |
| ConfGF | 88.49 | 94.13 | 0.2673 | 0.2685 | 62.15 | 70.93 | 1.1629 | 1.1596 |
| DGSM | 91.49 | 95.92 | 0.2139 | 0.2137 | 78.73 | 94.39 | 1.0154 | 0.9980 |
| GeoDiff | 90.54 | 94.61 | 0.2090 | 0.1988 | 89.13 | 97.88 | 0.8629 | 0.8529 |
| DMCG | **96.23** | **99.26** | **0.2083** | **0.2014** | **96.52** | **100.00** | **0.7220** | **0.7161** |
| Std | ±0.38 | ±0.37 | ±0.0052 | ±0.0040 | ±0.14 | ±0.00 | ±0.0027 | ±0.0061 |

| Methods | Large-scale QM9 | | | | Large-scale Drugs | | | |
| | COV(%)↑ | | MAT (Å)↓ | | COV(%)↑ | | MAT (Å)↓ | |
| | Mean | Median | Mean | Median | Mean | Median | Mean | Median |
|---|---|---|---|---|---|---|---|---|
| RDKit | 81.61 | 85.71 | 0.2643 | 0.2472 | 69.42 | 77.45 | 1.0880 | 1.0333 |
| CVGAE | 0.00 | 0.00 | 1.4687 | 1.3758 | 0.00 | 0.00 | 2.6501 | 2.5969 |
| GraphDG | 13.48 | 5.71 | 0.9511 | 0.9180 | 1.95 | 0.00 | 2.6133 | 2.6132 |
| CGCF | 81.48 | 86.95 | 0.3598 | 0.3684 | 57.47 | 62.09 | 1.2205 | 1.2003 |
| ConfVAE | 80.18 | 85.87 | 0.3684 | 0.3776 | 57.63 | 63.75 | 1.2125 | 1.1986 |
| ConfGF | 89.21 | 95.12 | 0.2809 | 0.2837 | 70.92 | 85.71 | 1.0940 | 1.0917 |
| GeoMol | 91.05 | 95.55 | 0.2970 | 0.2993 | 69.74 | 83.56 | 1.1110 | 1.0864 |
| DMCG | **98.34** | **100.00** | **0.1486** | **0.1340** | **96.22** | **100.00** | **0.6967** | **0.6552** |

conformations that are close to them within a predefined threshold. These results show the effectiveness and scalability of our method.

(2) For the molecules in GEOM-QM9 and GEOM-Drugs, our method achieves more improvement on molecules with more heavy atoms. Take the small-scale results in Table 1 as an example. On average, GEOM-QM9 and GEOM-Drugs have 8.8 and 24.9 heavy atoms respectively. In terms of MAT mean values, on GEOM-QM9, our method improves ConfGF and GeoDiff by 22.7% and 1.2%, while on GEOM-Drugs, the improvements are 37.9% and 16.3%. The results demonstrate the effectiveness of our method on large molecules.

More analysis is in Appendix B.5.

(3) Our method is much more sample-efficient than methods based on Langevin dynamics like ConfGF, since we can generate IID samples free of the auto-correlation in a Markov chain. ConfGF requires 5000 sequential forward steps, while we only need to sample once from $\mathcal{N}(0, \boldsymbol{I})$ and forward through the model. For a fair comparison, following the official implementation of ConfGF, we split the test sets of small-scale GEOM-QM9 and GEOM-Drugs into 200 batches. ConfGF requires 8511.60 and 11830.42 seconds to decode QM9 and Drugs test sets, while our method only requires 32.68 and 54.89 seconds respectively. Our method speeds up the decoding more than 200 times. Our method is also much more efficient than the recent GeoMol algorithm , which takes 99.34s and 668.95s to decode the above two datasets.

(4) As shown in Table 1, the standard derivations of our method are significantly smaller than the gain compared to the previously best results. This shows the effectiveness and robustness of our method. In

addition, considering that our method takes a random conformation as input, to test the confidence interval, we run decoding with 10 different initial conformations. The mean COV and MAT scores on small-scale GEOM-QM9 are $96.24 \pm 0.12$ and $0.2079 \pm 0.0010$, and those two numbers on small-scale GEOM-Drugs are $96.38 \pm 0.19$ and $0.7239 \pm 0.0025$. Our method is not sensitive to the choice of initial conformations.

The number of rotatable bonds is an important metric of how flexible a molecule is. We report coverage score w.r.t. the number of rotatable bonds in Figure 4 based on small-scale GEOM-Drugs. More rotatable bonds indicate harder generation. Our method outperforms previous baselines.

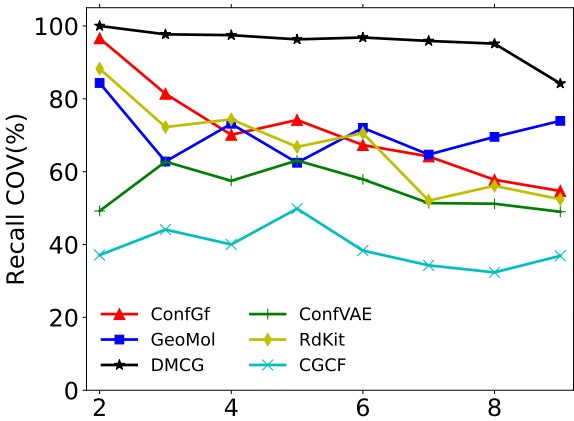

Figure 4: Coverage scores w.r.t. number of rotatable bonds.

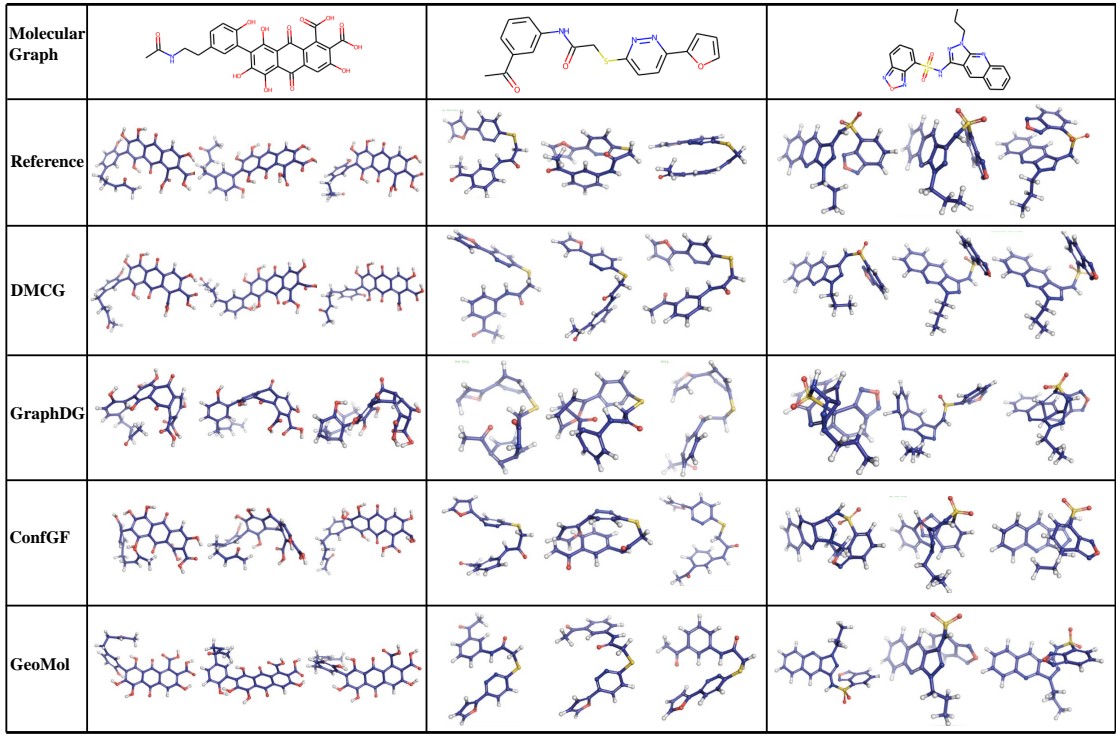

Figure 5: Visualization of different conformations.

In Figure 5, we visualize the conformation of different methods. We randomly select three molecules from the small-scale GEOM-drug dataset, generate several conformations, and visualize the best-aligned ones with the groundtruth. We can see that our method can generate high-quality conformations than previous methods, which are the most similar to the groundtruth.

**Computation cost analysis**: We use `PyTorch profiler`[3] to analyze the training time of the following components: (1) model forward time, which denotes the time of calculating the hidden representations from the input layer to output layer; (2) transformation time, which denotes the time of calculating the optimal roto-translation operation $\rho^*$; (3) permutation time, which denotes the time of enumerating all possible permutations in $\mathcal{S}$ and find the optimal one $\sigma^* \in \mathcal{S}$; note that we can use `torch.no_grad` to reduce time and memory; (4) loss forward time, which is the total of calculating the loss after obtaining $\rho^*$ and $\sigma^*$; (5) loss backward time, which denotes the time of gradient backpropagation.

The time is summarized in Table 2. We can see that model forward and loss forward/backward takes about 71.4% of the total computation time. The transformation and permutation takes 20.4% and 8.2% of the total time. Note that there are 7 transformation operations in the experiments (see Eqn.(8)). For the full training pipeline where data loading, model forwarding, loss forwarding, gradient backpropagation, metric calculation and CPU/GPU communications are all considered, DMCG takes 20% more time than that without roto-translation and permutation.

Table 2: Computation time statistics of each part in 100 iterations.

| Model forward | Transformation | Permutation | Loss forward | Loss backward |
|---|---|---|---|---|
| 5.515 (52.8%) | 2.136 (20.4%) | 0.858 (8.2%) | 0.052 (0.5%) | 1.886 (18.1%) |

We use graph isomorphism algorithms to find all $\mathcal{S}$. Although the general graph isomorphism problem is NP-hard, the size of drug-like molecules is largely limited, otherwise the molecule's druggability is limited (one can refer to Lipinski's rule of five). Therefore, our method does not need scalability to a large scale. In our experiments, it takes 4.9 seconds to process $10k$ molecules in GEOM-QM9, and 6.6 seconds to process $10k$ molecules in GEOM-Drugs. This is negligible compared to the training time, and we only need to process the data for one time in data preparing stage.

## 4.3 Molecular docking

Molecular docking (Roy et al., 2015) is a widely used technique in drug discovery, which aims to find the optimal binding conformation of a drug (i.e., the small molecule) in the pocket of a given target protein and the corresponding binding affinity. In most cases, the molecular docking algorithms treat proteins as rigid bodies and take one conformation of the small molecules as the initial structure inputs. The algorithms then search for the optimal conformation in the conformation space of the small molecules guided by the scoring function. However, due to the complexity of the conformation space, it is difficult for the algorithm to converge to a global minimum. Therefore, the choice of the initial structure often leads to different binding conformations and needs to be taken seriously.

Previously, RDKit was often used to generate initial conformations of small molecules, which usually got reasonable but not optimal results after docking. To verify the effectiveness of our method, we compared the docked poses which take initial conformations generated by our method (DMCG), ConfGF, GeoMol, GeoDiff and RDKit as the initial conformations for docking respectively.

We use Smina (Koes et al., 2013) for molecular docking and make evaluation on PDBbind refined set (Liu et al., 2017) which is a comprehensive collection of experimentally measured binding affinity for all biomolecular complexes deposited in the Protein Data Bank[4]. We randomly select 100 protein-ligand pairs for evaluation. Appendix A.3 shows detailed optimization hyper-parameters.

---

[3]`https://pytorch.org/tutorials/recipes/recipes/profiler_recipe.html`
[4]https://www.rcsb.org/

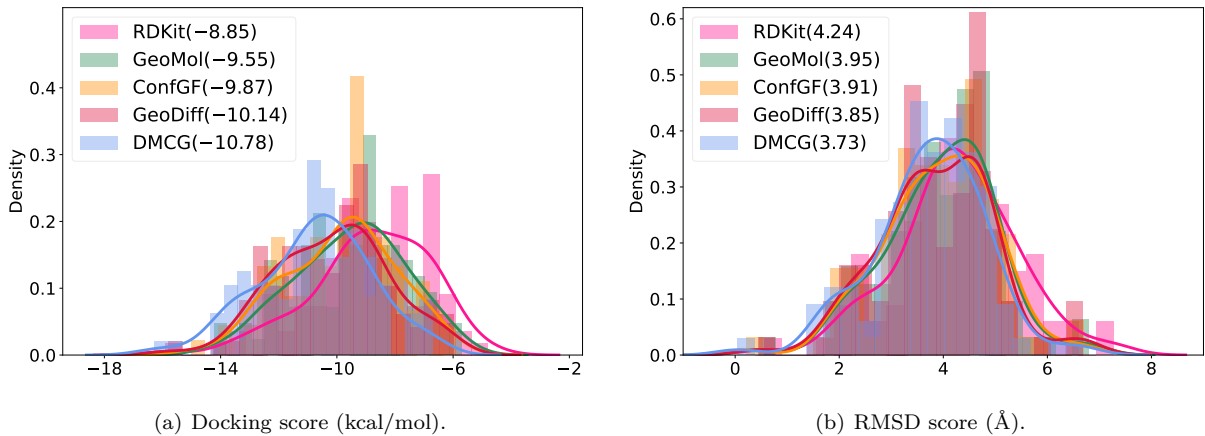

(a) Docking score (kcal/mol).

(b) RMSD score (Å).

Figure 6: Histograms of docking scores and RMSD scores.

Two metrics were used to evaluate the results of docking. One is the docking score (roughly, the estimation of binding affinity), which measures how well a molecule fits the binding site. A smaller value indicates better binding affinity. The other is the root-mean-square deviation (RMSD, the smaller, the better) compared to the crystal complex structure. As shown in Figure 6(a), the distributions for the three methods have the similar shape but our method is much more left-shifted than the others. This shows that for the same small molecule, our method tends to help docking to find conformations with higher binding affinities. Furthermore, docking tends to find lower RMSD binding conformations using the conformation generated by our method as the initial conformation, suggesting that our method can help docking to find binding conformations that are closer to the native crystal structures (Figure 6(b)). We also summarize the mean values of the docking scores and RMSD of different algorithms in the legends of Figure 6. All these results show that our method provides more proper initial conformations for molecular docking and thus facilitates the real application in computer-aided drug discovery.

### 4.4 Property prediction

In addition to conformation generation task, we also conduct experiments on property prediction task, which is to predict molecular property based on an ensemble of generated conformation (Axelrod & Gomez-Bombarelli, 2021). We first randomly choose 30 molecules from GEOM-QM9 test sets, and then sample 50 conformations for each molecule using RDKit, ConfGF and our method. We use the quantum chemical calculation package Psi4 (Smith et al., 2020) to calculate the energy, HOMO and LUMO for each generated conformation and groundtruth conformation. Next, we calculate the ensemble properties of average energy $\overline{E}$, lowest energy $E_{\min}$, average HOMO-LUMO gap $\overline{\Delta\epsilon}$, minimum gap $\Delta\epsilon_{\min}$ and maximum gap $\Delta\epsilon_{\max}$ based on the conformational properties of each molecule[5]. We use mean absolute error to measure the property differences between the generated conformations and groundtruth conformations.

The results are shown in Table 3. Our method significantly outperforms GraphDG, ConfGF and the recent GeoMol, which shows the effectiveness of our method. We can observe that RDKit achieves the best results on $\Delta\epsilon_{\min}$, and we will combine our method with RDKit in the future.

### 4.5 Ablation study

We conduct ablation study on the small-scale GEOM-Drugs dataset. The results are shown in Table 4.

---

[5]From a physics perspective, using the Boltzmann-weighted average of the energies of the molecules is a better choice, but the distribution is missing from the dataset. Following (Simm & Hernández-Lobato, 2020; Shi et al., 2021; Luo et al., 2021b), we use the average number here instead of the weighted version.

Table 3: Mean absolute error of predicted ensemble properties. (Unit: eV).

| Methods | $\overline{E}$ | $E_{\min}$ | $\overline{\Delta\epsilon}$ | $\Delta\epsilon_{\min}$ | $\Delta\epsilon_{\max}$ |
|---------|------|-----------|-----------|-----------------|-----------------|
| RDKit | 0.8875 | 0.6530 | 0.3484 | **0.5570** | 0.2399 |
| GraphDG | 45.1088 | 9.2868 | 3.8970 | 6.6997 | 1.7724 |
| ConfGF | 2.8349 | 0.2012 | 0.6903 | 4.9221 | 0.1820 |
| GeoMol | 4.5700 | 0.5096 | 0.5616 | 3.5083 | 0.2650 |
| DMCG | **0.4324** | **0.1364** | **0.2057** | 1.3229 | **0.1509** |

Table 4: Ablation study on small-scale GEOM-Drugs.

| Methods | COV(%)↑ | | MAT (Å)↓ | |
|---------|---------|--------|----------|--------|
|  | Mean | Median | Mean | Median |
| DMCG | 96.52 | 100.00 | 0.7220 | 0.7161 |
| No $\ell_{\mathrm{P}}$ | 77.78 | 86.09 | 1.0657 | 1.0563 |
| No attention | 94.99 | 100.00 | 0.7611 | 0.7581 |
| No normalization | 92.77 | 98.68 | 0.8002 | 0.7977 |

(1) We remove the permutation invariant loss and use the roto-translation invariant loss only, i.e., the $\ell_{\mathrm{RTP}}$ in Eqn.(2) is replaced with $\ell_{\mathrm{RT}}$ defined in Section.(2.1). The results are denoted as "No $\ell_{\mathrm{P}}$" in Table 4.

(2) We replace attentive node aggregation by a simple `MLP` network. That is, Eqn.(5) is replaced by

$$h_i^{(l)} = h_i^{(l-1)} + \mathtt{MLP}(h_i^{(l-1)}, U^{(l-1)}, \frac{1}{|N(i)|} \sum_{j \in N(i)} h_j^{(l-1)}).$$

The results are denoted as "No attention" in Table 4.

(3) We remove the normalization step in Eqn.(7), i.e., the $m^{(l)}$ is not used. Denote the results as "No normalization".

We can see that: (1) The permutation invariant loss is extremely important, without which the mean COV drops 18.91 while MAT increases 0.3434. We also visualize several cases in Appendix B.2 to compare the results with or without $\ell_{\mathrm{P}}$. (2) Without attentively aggregating the atom features, the mean COV drops 1.70 points and MAT score increases 0.0345 points. (3) Without the conformation normalization, the performance is also hurt. These results demonstrate the importance of the components in our method.

Finally, we compute the COV and MAT scores of $\hat{R}^{(l)}$ against the groundtruth, which is the output conformation of the $l$-th block in the decoder. $\hat{R}^{(0)}$ is the output of $\varphi_{2\mathrm{D}}$. The results are shown in Figure 7. We can see that iteratively refining the conformations can improve the performances, which shows the effectiveness of our design. This phenomenon is consistency with the discovery in machine translation (Xia et al., 2017), image synthesis (Chen & Koltun, 2017) and protein structure prediction (Jumper et al., 2021).

We leave the discussion about additional constraints on loss functions, the comparison of model sizes and more discussions in Appendix B.

## 5 Conclusions and future work

In this work, we propose a new method, that directly generates the coordinates of conformations. For this purpose, we design a dedicated loss function, which is invariant to roto-translation and permutation on symmetric atoms. We also design a new model with many advanced modules (i.e., GATv2, GN block) that can iteratively refine the conformations. Experimental results on both small-scale and large-scale GEOM-QM9 and GEOM-Drugs demonstrate the effectiveness of our method.

For future work, first, we will incorporate chemical rules into deep learning models to improve generation quality. Second, current methods are mainly non-autoregressive, where all coordinates are generated simul-

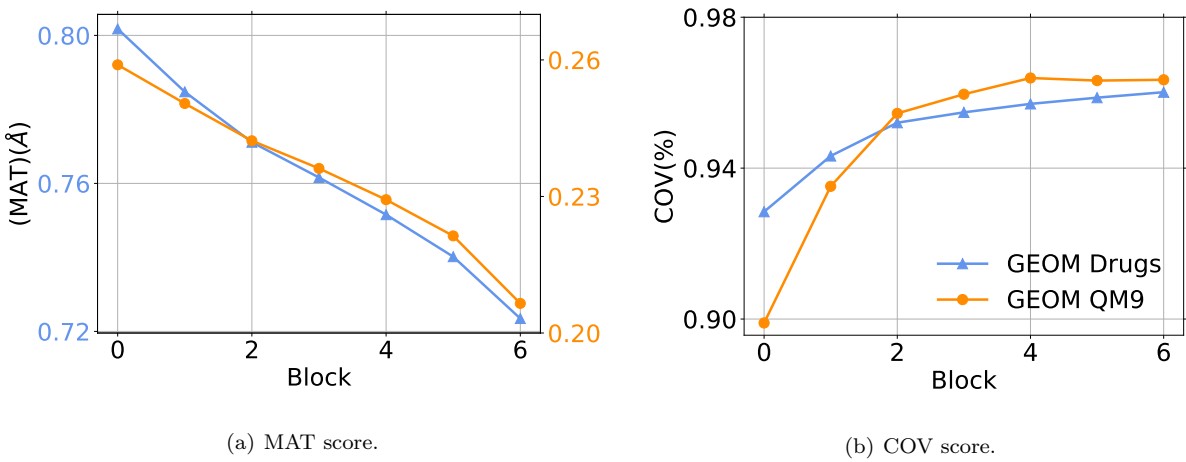

(a) MAT score.

(b) COV score.

Figure 7: The MAT and COV scores of $\hat{R}^{(l)}$ output by different blocks.

taneously. We will study the autoregressive setting so as to further improve the accuracy. Third, Villar et al. (2021) point that equivariance/invariance can be universally approximated through polynomial functions. This is a good direction to explore in molecular conformation generation. Fourth, when the number of permutation invariant mappings in a molecule is extremely large, enumerating all of them is not the best choice due to the exponentially increased computation cost. We will improve our method along this direction. Fifth, we will deeply collaborate with chemists and biologists on more case studies.

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

## A  Procedure descriptions

### A.1  Details of other model components

The model architectures of $\varphi_{2D}$ and $\varphi_{3D}$ are similar to $\varphi_{dec}$, with the following differences.

Comparing $\varphi_{2D}$ with $\varphi_{dec}$, the differences are the initial conformation $\hat{R}^{(0)}$ and initial features (i.e., the $H_V^{(0)}$, $H_E^{(0)}$ and $U^{(0)}$). $\varphi_{2D}$ takes a random conformation sampled from uniform distribution in $[-1, 1]$ as input. The initial atom and edge features are the embeddings of the atoms and edges respectively. $\varphi_{2D}$ will also output a prediction of the conformation. Note that the random variable $z$ sampled from Gaussian $\mathcal{N}(\mu_{R,G}, \Sigma_{R,G})$ is not used in $\varphi_{2D}$.

Comparing $\varphi_{3D}$ with $\varphi_{dec}$, the differences are the initial conformation $\hat{R}^{(0)}$, initial features (i.e., the $H_V^{(0)}$, $H_E^{(0)}$ and $U^{(0)}$) too. $\varphi_{3D}$ takes the groundtruth conformation as input. The initial atom and edge features are the embeddings of the atoms and edges respectively. Another difference is that the fourth step of $\varphi_{dec}$, i.e., Eqn.(7), is not used.

### A.2 More details about training

We use AdamW optimizer (Loshchilov & Hutter, 2019) with initial learning rate $\eta_0 = 2 \times 10^{-4}$ and weight decay 0.01. In the first 4000 iterations, the learning rate is linearly increased from $10^{-6}$ to $2 \times 10^{-4}$. After that, we use cosine learning rate scheduler (Loshchilov & Hutter, 2016), where the learning rate at the $t$-th iteration is $\eta_0(1 + \cos(\pi \frac{t}{T}))/2$, where $T$ is the half of the period (i.e., the iteration numbers of 10 epochs in our setting). Similarly, we also use the cosine scheduler to dynamically set the $\beta$ at range $[0.0001, \beta_{\max}]$. The batch size is fixed as 128. All models are trained for 100 epochs. For the two small-scale settings, the experiments are conducted on a single V100 GPU. For the two large-scale settings, we use two V100 GPUs for experiments. The $\lambda$ in Eqn.(8) for large-scale QM9 is 0.1, and for the remaining settings, $\lambda$ is set as 0.2. The detailed hyper-parameters are described in Table 5. We grid search the best hyper-parameter with these hyper-parameters, and the last hyper-parameters are selected according to validation performance, *i.e.*, the hyper-parameter setting corresponding to the best coverage score (COV) and matching score (MAT) on the hold-out validation set.

Table 5: Hyper-parameters for our experiments.

|  | Small-Scale | Large-Scale |
| --- | --- | --- |
| Layer number | 6 | 6 |
| Dropout | {0.1, 0.2} | {0.1, 0.2} |
| Learning rate | {1e-4, 2e-4, 5e-4} | {1e-4, 2e-4, 5e-4} |
| Batch size | 128 | 128 |
| Epoch | 100 | 100 |
| $\beta$ Min | 0.0001 | 0.001 |
| $\beta$ Max | {0.001, 0.002, 0.004, 0.008, 0.01} | {0.005, 0.01, 0.02, 0.04, 0.05} |
| Latent size | 256 | 256 |
| Hidden dimension | 1024 | 1024 |
| GPU number | 1× NVIDIA V100 | 2× NVIDIA V100 |

### A.3 More details about molecular docking

For RDKit, we generated one initial conformation as input and set *num_modes* to 50 when performing docking[6]. For our method, ConfGF, GeoDiff and GeoMol, since the generated conformations are independent and diverse, we randomly selected five of them, performed five independent molecular docking calculations and set *num_modes* to 10 to ensure all three methods generate equal number of conformations. Eventually, each method got about 50 binding conformations. The conformation corresponding to the lowest binding affinity was selected as the final docked pose.

## B More experimental results

### B.1 Precision-based results

The precision-based coverage and matching scores are defined as follows:

$$
\text{COV-P}(\mathbb{S}_g, \mathbb{S}_r) = \frac{1}{|\mathbb{S}_g|} \left| \left\{ \hat{R} \in \mathbb{S}_g \mid \text{RMSD}(R, \hat{R}) < \delta, \exists R \in \mathbb{S}_r \right\} \right|;
$$

$$
\text{MAT-P}(\mathbb{S}_g, \mathbb{S}_r) = \frac{1}{|\mathbb{S}_g|} \sum_{\hat{R} \in \mathbb{S}_g} \min_{R \in \mathbb{S}_r} \text{RMSD}(R, \hat{R}).
$$

(10)

The results are in Table 6. The results of GraphDG, CGCF, ConfVAE, ConfGF and GeoDiff are from (Xu et al., 2022). Our method is still the best one.

---

[6]When using different random seeds, the conformations output by RDKit is not diverse enough. Therefore, we only choose one here.

Table 6: Precision-based coverage and matching scores. Bold fonts indicate the best results.

| | Small-scale QM9 | | | | Small-scale Drugs | | | |
| | COV-P(%)↑ | | MAT-P(Å)↓ | | COV-P(%)↑ | | MAT-P(Å)↓ | |
| Methods | Mean | Median | Mean | Median | Mean | Median | Mean | Median |
|---|---|---|---|---|---|---|---|---|
| GraphDG | 43.90 | 35.33 | 0.5809 | 0.5823 | 2.08 | 0.00 | 2.4340 | 2.4100 |
| CGCF | 36.49 | 33.57 | 0.6615 | 0.6427 | 21.68 | 13.72 | 1.8571 | 1.8066 |
| ConfVAE | 38.02 | 34.67 | 0.6215 | 0.6091 | 22.96 | 14.05 | 1.8287 | 1.8159 |
| ConfGF | 49.02 | 46.69 | 0.5111 | 0.4979 | 23.15 | 15.73 | 1.7304 | 1.7106 |
| GeoDiff | 52.79 | 50.29 | 0.4448 | 0.4267 | 61.47 | 64.55 | 1.1712 | 1.1232 |
| GeoMol | 84.98 | 89.90 | 0.3292 | 0.3269 | 75.54 | 94.13 | 1.0028 | 0.9082 |
| DMCG | **87.26** | **91.00** | **0.2872** | **0.2926** | **81.05** | **95.51** | **0.9210** | **0.8785** |
| Dataset | Large-scale QM9 | | | | Large-scale Drugs | | | |
| Methods | Mean | Median | Mean | Median | Mean | Median | Mean | Median |
| | COV-P(%)↑ | | MAT-P(Å)↓ | | COV-P(%)↑ | | MAT-P(Å)↓ | |
| ConfGF | 46.23 | 44.87 | 0.5171 | 0.5133 | 28.23 | 20.71 | 1.6317 | 1.6155 |
| GeoMol | 78.28 | 81.03 | 0.3790 | 0.3861 | 41.46 | 36.79 | 1.5120 | 1.5107 |
| DMCG | **90.86** | **95.36** | **0.2305** | **0.2258** | **74.57** | **81.80** | **0.9940** | **0.9454** |

## B.2 Combination with distance-based and angle-based loss functions

One may be curious about whether using distance-based loss and angle-based can further improve the performance, since the latter two are equivariant to the transformation of coordinates. For ease of reference, let $R_i$ denote the groundtruth coordinate of atom $v_i$ and $\hat{R}_i$ denote the predicted coordinate of atom $v_i$. Recall in Section 1, we use $E$ to denote the collection of all bonds. We define $E_2$ as $\{(i, j, k)|(i, j) \in E, (i, k) \in E, k \neq j\}$.

Inspired by (Winter et al., 2021) and (Ganea et al., 2021), we use the following two functions:

$$\ell_{\text{angle}} = \frac{1}{|E_2|} \sum_{(i,j,k) \in E_2,} \| \text{cosine}(R_j - R_i, R_k - R_i) - \text{cosine}(\hat{R}_j - \hat{R}_i, \hat{R}_k - \hat{R}_i)\|_F^2, \tag{11}$$

$$\ell_{\text{bond}} = \frac{1}{|E|} \sum_{(i,j) \in E} \left( \text{distance}(R_j, R_i) - \text{distance}(\hat{R}_j - \hat{R}_i) \right)^2, \tag{12}$$

where $\text{cosine}(a, b) = \frac{a^\top b}{\|a\|\|b\|}$ and $\text{distance}(a, b) = \|a - b\|$, $a$ and $b$ are two vectors. That is, we apply additional constraints to bond length and bond angles. Please note that with the above two auxiliary loss functions, our method still generates coordinates directly and does not need to generate intermediate distances and angles.

We verify the following three loss functions:

$$\mathcal{L}_1 = \mathbb{E}_{\epsilon \sim \mathcal{N}(0,I)} \ell_{\text{RT}}(R, \hat{R}(\mu_{R,G} + \Sigma_{R,G}\epsilon, G)) + \beta D_{\text{KL}}(\mathcal{N}(\mu_{R,G}, \Sigma_{R,G})\|\mathcal{N}(0, I)), \tag{13}$$

$$\mathcal{L}_2 = \mathbb{E}_{\epsilon \sim \mathcal{N}(0,I)} \ell_{\text{RT}}(R, \hat{R}(\mu_{R,G} + \Sigma_{R,G}\epsilon, G)) + \beta D_{\text{KL}}(\mathcal{N}(\mu_{R,G}, \Sigma_{R,G})\|\mathcal{N}(0, I)) + \lambda_1(\ell_{\text{angle}} + \ell_{\text{bond}}), \tag{14}$$

$$\mathcal{L}_3 = \mathbb{E}_{\epsilon \sim \mathcal{N}(0,I)} \ell_{\text{RTP}}(R, \hat{R}(\mu_{R,G} + \Sigma_{R,G}\epsilon, G)) + \beta D_{\text{KL}}(\mathcal{N}(\mu_{R,G}, \Sigma_{R,G})\|\mathcal{N}(0, I)) + \lambda_1(\ell_{\text{angle}} + \ell_{\text{bond}}), \tag{15}$$

where $\lambda_1 = 0.1$. Note in Eqn.(13) and Eqn.(14), we use the roto-translation loss only without considering permutation invariant loss on symmetric atoms. We conduct experiments on GEOM-Drugs (small-scale setting). The results are reported in Table 7.

We have the following observations:

(1) Comparing $\mathcal{L}_1$ with our method, we can see that using permutation invariant loss on symmetric atoms are important, without which the results significantly drop. (2) Comparing $\mathcal{L}_2$ with our method, we can see that

Table 7: Results of combining with constraints on bond lengths and bond angles.

| Methods | COV(%)↑ | | MAT (Å)↓ | |
|---|---|---|---|---|
| | Mean | Median | Mean | Median |
| DMCG | **96.52** | **100.00** | **0.7220** | **0.7161** |
| $\mathcal{L}_1$ | 77.78 | 86.09 | 1.0657 | 1.0563 |
| $\mathcal{L}_2$ | 92.45 | 98.70 | 0.8983 | 0.9016 |
| $\mathcal{L}_3$ | 96.01 | 100.00 | 0.7235 | 0.7199 |

when we do not use the permutation invariant loss, using more constraints on bond lengths and bond angles can help improve the performances. (3) When using both permutation invariant loss and roto-translation invariant loss, using $\ell_{\text{bond}}$ and $\ell_{\text{angle}}$ will not bring more significant improvement. These results demonstrate that for molecular conformation generation, it is important to consider the permutation of symmetric atoms.

To illustrate the impact of the permutation invariant loss, we show two examples in Figure 8. For these two examples, there exists a rotatable ring at the end of a molecule, where the ring is symmetric to the bond connecting itself to the rest of the molecule. Without the permutation invariant loss (see the row No $\ell_P$), our method fails to generate the coordinates of such rings, but simply puts them in a line. This is because the model is trapped into local optimal. By using the permutation invariant loss, we can successfully recover the conformations of those rings (see the row "DMCG"). This shows the importance of using the permutation invariant loss $\ell_P$ as we proposed.

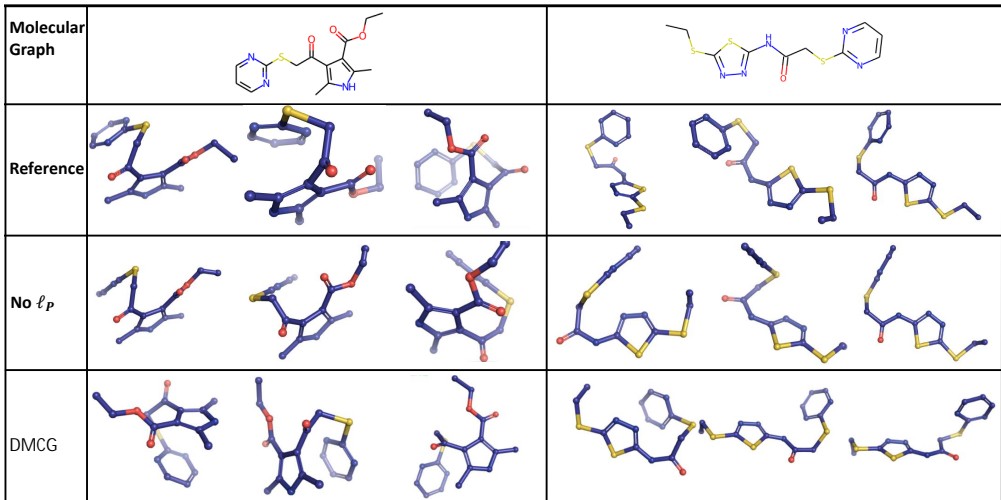

Figure 8: The illustration of the impact of the permutation invariant loss. "No $\ell_P$" means without the permutation invariant loss.

### B.3 Gradient back-propagation through roto-translational operation?

As introduced Section 2.1, the optimal roto-translation operation $\rho^*$ can be obtained by calculating the eigenvalues and eigen vectors of a matrix. This is implemented by using the `torch.linalg.eig`. However, the official document lists a warning of this function: "Gradients computed using the eigenvectors tensor will only be finite when $A$ has distinct eigenvalues. Furthermore, if the distance between any two eigenvalues is close to zero, the gradient will be numerically unstable, as it depends on the eigenvalues $\lambda_i$ through the computation of $\frac{1}{\min_{i \neq j} \lambda_i - \lambda_j}$" (the words are from the official document). Therefore, we disable the gradients through $\rho^*$ for stability.

We compare the performance between enabling and disabling the gradients. The results are in Table 8. Overall speaking, enabling the gradients slightly hurts the performance (especially the MAT for GEOM-Drugs) and increases computation time. Therefore, we recommend disabling the gradients through $\rho^*$.

Table 8: Results with/without gradient back-propagation through $\rho^*$.

| | Small-scale QM9 | | | | Small-scale Drugs | | | |
| | COV-P(%)↑ | | MAT-P(Å)↓ | | COV-P(%)↑ | | MAT-P(Å)↓ | |
| Methods | Mean | Median | Mean | Median | Mean | Median | Mean | Median |
|---|---|---|---|---|---|---|---|---|
| with gradient | 96.14 | 99.43 | 0.2090 | 0.2062 | 96.08 | 100.00 | 0.7345 | 0.7247 |
| without gradient | 96.23 | 99.26 | 0.2083 | 0.2014 | 96.52 | 100.00 | 0.7220 | 0.7161 |

## B.4 Study of model parameters

In this section, we compare the performances of our method and ConfGF. The ConfGF model has 0.81M parameters. We reduce the network parameter of our method to $0.98M$. The results are shown in Table 9.

Table 9: Comparison of our method and ConfGF with different model sizes

| Dataset | GEOM-QM9 | | | | GEOM-Drugs | | | |
| Methods | COV(%)↑ | | MAT (Å)↓ | | COV(%)↑ | | MAT (Å)↓ | |
| | Mean | Median | Mean | Median | Mean | Median | Mean | Median |
|---|---|---|---|---|---|---|---|---|
| ConfGF (0.81M) | 88.49 | 94.13 | 0.2673 | 0.2685 | 62.15 | 70.93 | 1.1629. | 1.1596 |
| DMCG (0.98M) | 94.28 | 98.20 | 0.2399 | 0.2361 | 89.40 | 97.06 | 0.8653 | 0.8670 |
| DMCG (normal) | 96.23 | 99.26 | 0.2083 | 0.2014 | 96.52 | 100.00 | 0.7220 | 0.7161 |

By reducing the network parameters of our method, the performance also drops, but still significantly better than ConfGF.

We also study the results of our method w.r.t. the representation dimension $d$ (please kindly refer to the Section 2.2) and the dimension of the MLP layer (denoted as $d_{\text{MLP}}$). Results are reported in Table 10. We can see that our model benefits from more parameters.

Table 10: Results on small-scale GEOM-Drugs with different hidden dimensions.

| | COV(%)↑ | | MAT (Å)↓ | |
| | Mean | Median | Mean | Median |
|---|---|---|---|---|
| $(128, 512)$ | 95.54 | 100.00 | 0.7641 | 0.7584 |
| $(128, 1024)$ | 95.84 | 100.00 | 0.7514 | 0.7432 |
| $(256, 512)$ | 95.88 | 100.00 | 0.7378 | 0.7387 |
| $(256, 1024)$ | 96.52 | 100.00 | 0.7220 | 0.7161 |

## B.5 More discussions on the conformation with more heavy atoms

In Table 1, we observe that our method works better than distance-based methods (include modeling the distances directly, or the gradients of distances) on molecules with more heavy atoms. Our conjecture is that for these distance-based works, they usually extend the molecular graph with 1,2,3-order neighbors, which is sufficient to determine the 3D structure in principle. For GEOM-QM9 dataset, considering the number of atoms is less than 10, this extended graph is nearly a complete graph and can provide enough signals to reconstruct the 3D structure. Therefore, these distance-based performances are good on GEOM-QM9 dataset. For GEOM-Drugs dataset, the numbers of atoms are much more than those in GEOM-QM9. Although in theory, the distances in a third-order extended graph can reconstruct the 3D structure,

practically the signals are still not enough. Our method does not rely on the interatomic distances, and can achieve good results on large molecules.

To verify our conjecture, on GEOM-Drugs, we categorize the molecules based on their numbers of heavy atoms. We choose one of the five independently run DMCG models for analysis. The number of heavy atoms in the $i$-th group lie in $[10i + 1, 10(i + 1)]$. We compare our method against ConfGF (the code of DGSM is not available) and GraphDG. The results are in Table 11. We have similar observation, that our method brings more improvements than previous method on larger molecules.

Table 11: COV and MAT mean scores w.r.t numbers of heavy atoms on small-scale GEOM-Drugs. The $i$ indicates that the heavy atom number lies in range $[10i + 1, 10(i + 1)]$, $i \in \{1, 2, 3\}$.

| Metric | COV(%)↑ | | | | MAT(Å)↓ | | | |
|---|---|---|---|---|---|---|---|---|
| | $i = 1$ | $i = 2$ | $i = 3$ | average | $i = 1$ | $i = 2$ | $i = 3$ | average |
| ConfGF | 99.95 | 66.28 | 15.34 | 62.54 | 0.7764 | 1.1510 | 1.5345 | 1.1637 |
| GraphDG | 15.11 | 1.78 | 0.0 | 3.12 | 2.0578 | 2.5863 | 2.9849 | 2.5847 |
| DMCG | **100.00** | **97.62** | **90.04** | **96.69** | **0.5305** | **0.7190** | **0.8794** | **0.7223** |

## B.6 More results about property prediction

Table 12: Median absolute error of predicted ensemble properties. (Unit: eV).

| Methods | $\overline{E}$ | $E_{\min}$ | $\overline{\Delta\epsilon}$ | $\Delta\epsilon_{\min}$ | $\Delta\epsilon_{\max}$ |
|---|---|---|---|---|---|
| RDKit | 0.8721 | 0.6119 | 0.3057 | **0.4414** | 0.1830 |
| GraphDG | 13.1707 | 1.9221 | 3.4136 | 7.6845 | 1.1663 |
| ConfGF | 1.5167 | 0.1972 | 0.6588 | 4.8920 | 0.1686 |
| DMCG | **0.4132** | **0.1100** | **0.1276** | 0.8486 | **0.1288** |

The median absolute error of the property prediction is shown in Table 12. We can see that our method still outperforms all deep learning based methods, which demonstrate the effectiveness of our method.

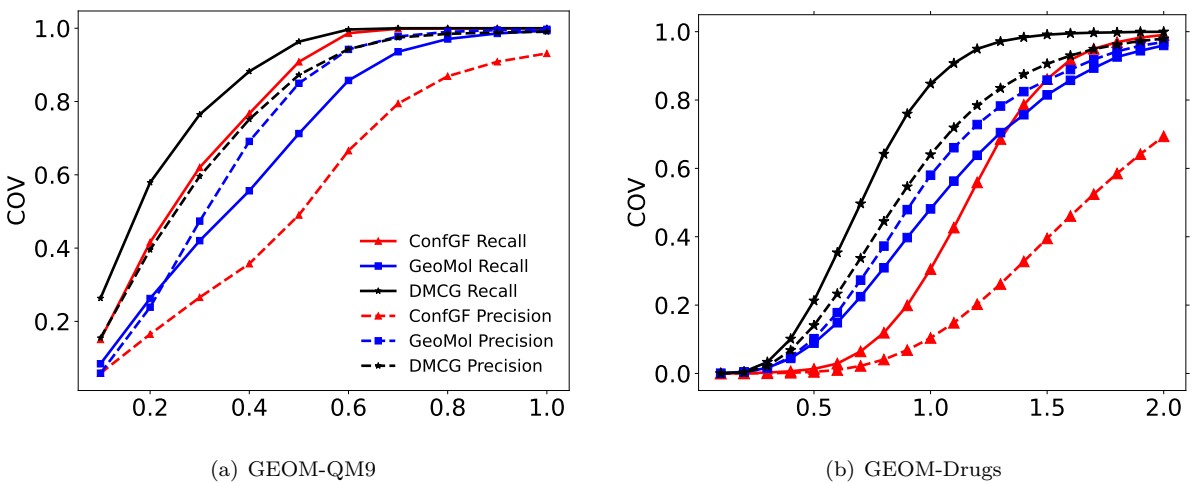

(a) GEOM-QM9

(b) GEOM-Drugs

Figure 9: The Coverage score w.r.t different threshold $\delta$.

## B.7 Results with different training sizes

To investigate whether our method relies on a large dataset, we subsample training data of the large-scale GEOM-QM9 and GEOM-Drugs to 10%, 25%, 50%, 75%. The validation and test sets remain unchanged. The results are in Table 13.

Table 13: Results with different training sizes. The baseline mehtods are trained on the full dataset. For our DMCG, it is trained on 10%, 25%, 50%, 75% and the full dataset.

| Dataset | GEOM-QM9 | | | | GEOM-Drugs | | | |
|---|---|---|---|---|---|---|---|---|
| | COV(%)↑ | | MAT (Å)↓ | | COV(%)↑ | | MAT (Å)↓ | |
| Methods | Mean | Median | Mean | Median | Mean | Median | Mean | Median |
| ConfGF | 89.21 | 95.12 | 0.2809 | 0.2837 | 70.92 | 85.71 | 1.0940 | 1.0917 |
| GeoMol | 91.05 | 95.55 | 0.2970 | 0.2993 | 69.74 | 83.56 | 1.1110 | 1.0864 |
| DMCG (10%) | 91.48 | 97.45 | 0.2748 | 0.2692 | 94.47 | 100.00 | 0.7934 | 0.7624 |
| DMCG (25%) | 97.65 | 100.00 | 0.1858 | 0.1676 | 95.17 | 100.00 | 0.7475 | 0.7092 |
| DMCG (50%) | 98.23 | 100.00 | 0.1606 | 0.1457 | 96.38 | 100.00 | 0.7057 | 0.6771 |
| DMCG (75%) | 98.31 | 100.00 | 0.1544 | 0.1384 | 96.14 | 100.00 | 0.6947 | 0.6562 |
| DMCG (100%) | 98.34 | 100.00 | 0.1486 | 0.1340 | 96.22 | 100.00 | 0.6967 | 0.6552 |

We can see that:

1. Generally, DMCG benefits from more training data.

2. With 10% training data, our method is better than previous baselines ConfGF and GeoMol.

## B.8 Adding more blocks

In this section, we study whether adding more blocks are helpful. We increase the number of blocks from 6 to 12. The results are in Table 14. For GEOM-QM9, we do not observer performance improvement by increasing the number of blocks. For GEOM-Drugs, increasing the number of blocks further improves the performance. Our conjecture is that, the molecules in GEOM-Drugs are more complex than those in GEOM-QM9, which benefit more from larger models.

Table 14: Results with different number of blocks.

| Dataset | GEOM-QM9 | | | | GEOM-Drugs | | | |
|---|---|---|---|---|---|---|---|---|
| | COV(%)↑ | | MAT (Å)↓ | | COV(%)↑ | | MAT (Å)↓ | |
| # blocks | Mean | Median | Mean | Median | Mean | Median | Mean | Median |
| 6 | 96.23 | 99.26 | 0.2083 | 0.2014 | 96.52 | 100.00 | 0.7220 | 0.7161 |
| 8 | 95.55 | 98.91 | 0.2217 | 0.2190 | 96.77 | 100.00 | 0.7122 | 0.7093 |
| 10 | 95.71 | 99.52 | 0.2215 | 0.2212 | 97.29 | 100.00 | 0.7089 | 0.7079 |
| 12 | 94.65 | 99.11 | 0.2280 | 0.2284 | 97.11 | 100.00 | 0.7092 | 0.6996 |

## B.9 Iterative refinement v.s. recursive refinement

Currently, the parameters of the blocks in the decoder (i.e., $\varphi_{\text{dec}}$) are not shared. Another option is to implement a recursive model, where the parameters of different decoder blocks are shared. The results are in Table 15. We can see that using the recursive model hurts the performances.

Table 15: Results with recursive decoder.

| Dataset | GEOM-QM9 | | | | GEOM-Drugs | | | |
|---|---|---|---|---|---|---|---|---|
| | COV(%)↑ | | MAT (Å)↓ | | COV(%)↑ | | MAT (Å)↓ | |
| # Method | Mean | Median | Mean | Median | Mean | Median | Mean | Median |
| DMCG | 96.23 | 99.26 | 0.2083 | 0.2014 | 96.52 | 100.00 | 0.7220 | 0.7161 |
| DMCG with recursive decoder | 94.03 | 98.00 | 0.2726 | 0.2771 | 95.20 | 100.00 | 0.7883 | 0.7862 |

## C Constraints on distances

Let $G$ be a molecular graph with $N$ atoms ($N \geq 3$). Let $d_{ij}$ denote the distance between atom $i$ and atom $j$. Define $D$ as the distance matrix, which is an $N \times N$ matrix, and $d_{ij}$ locates in the $i$-th row and $j$-th column of $D$.

The triangle inequalities means that for any three different $i, j, k$, $d_{i,j} + d_{j,k} \geq d_{j,k}$.

A valid distance matrix $D$ is induced from the $3N-6$ degree-of-freedom (DOF) of $N$ 3D-coordinates excluding global translation and rotation, while the popular practice of independently generating distances to 2- or 3-hop neighbors (Xu et al., 2021a) often introduces more DOF.

Moreover, a distance matrix should have a rank at most 5 after element-wise squared (Dokmanic et al., 2015). In other words, the rank of matrix $\tilde{D} = \{d_{ij}^2\}_{i,j}$ is at most 5. Such a constraint is hard to guarantee even if the DOF is matched (Simm & Hernández-Lobato, 2020) (e.g., $\lambda I$ has one DOF but is almost surely full-rank). It also makes gradients ill-defined (Shi et al., 2021) (other distances cannot all be held constant while taking an infinitesimal change to $d_{ij}$). Careful treatments (Hoffmann & Noé, 2019) often increase the order of computation complexity.

