# OpenReview forum: "Direct Molecular Conformation Generation"
_TMLR — Accepted by TMLR_

### Review · Reviewer_PiFm · 2022-07-13

**Summary Of Contributions:**

This work proposes a method to generate 3D molecular structures called "Direct Molecular Conformation Generation" (DMCG). The method is a latent variable model which generates a conformer from the output of a trained graph neural network and a random latent variable. The model is trained using an ELBO-like objective where the latent variable is outputted by a learned 3D encoder which is regularized by a KL divergence term. The authors perform a variety of experiments on different conformer generation benchmarks and report promising results.

The main contributions of the work are:
1. Formulating a loss which is invariant to permutations, rotations, and translations, so that the model is only penalized for generating an incorrect structure, not generating a correct structure in the wrong orientation/position/arbitrary breaking of permutation symmetry.
1. A model architecture inspired by AlphaFold which generates a structure and iteratively refines it.
1. Thorough empirical comparison of DMCG with other methods, both on standard benchmarks and on a more real-world docking and quantum chemical modelling task.

Overall I think this is a fairly strong paper. There is a clear contribution, and the claims are well-supported by the experiments. I believe it will be of interest to the TMLR audience. I have some small concerns about correctness but I imagine that these will be resolved without issue.

**Broader Impact Concerns:**

No concerns about ethical implications of the work.

**Requested Changes:**

### Very important changes

- Clarification of whether the algorithm to determine molecular symmetries described in section a.1 is incorrect, or whether I have misunderstood it (see "Weaknesses" above). If I have misunderstood it, then I suggest that the authors provide a clearer description.
- Clarification of experimental details from the "weaknesses" section.

### Nice changes

- Remove "space limitation" stuff and move appendix A into the main text
- You use the phrase "expressiveness of our loss function": what does this mean? I don't think that losses "express" anything... I would like to see this re-phrased to be clearer.
- The equations for sampling from a standard normal distribution on page 3 are only correct if $\Sigma$ is diagonal and if it is interpreted as standard deviation instead of variance (which is more common). This should be updated/clarified.
- DMCG (and other similar methods) depend on having a large dataset of high-quality conformers, which is often unavailable in practice. It would be very good to comment on how the method's performance changes with dataset size (perhaps an additional experiment), or at the very least discuss this limitation explicitly in the paper.

### Typos/other small things

- "we compares" -> "we compared" on page 9
- You never explicitly define the acronym DMCG; it just appears in your github link and in the legend of Figure 4. You should really change this.
- In equation 2, it would be nice to see explicitly which variables the minimization is taken with. It was slightly unclear to me.

**Strengths And Weaknesses:**

### Strengths

- **Interesting and principled loss function**: which empirically leads to much better performance (section 4.5). I am surprised that other works have not used such a loss function before (although I am not super familiar with work in this area so I could be wrong).
- **Strong empirical results on standard benchmarks**, also featuring numerical improvements over previous methods. The authors also compared to an impressive number of baselines!
- **Additional insightful experiments** including an ablation study on the key features of their method, and a docking/HOMO-LUMO task. I thought these experiments were interesting and insightful, particularly the ablation study.
- **Writing** is very clear and high-quality with few exceptions

### Weaknesses

- **Potential poor scaling of loss not discussed**: my understanding is that evaluating equation (1) requires enumerating all possible permutations, and for each permutation finding the best roto-translation analytically. However, for many larger molecules the number of possible permutations will be very high. For example, many amino acids (e.g. Tyrosine, Leucine, Valine; see https://en.wikipedia.org/wiki/Amino_acid) have at least 1 symmetric atom in their side chains. A peptide of $N$ amino acids will therefore have at least $2^N$ different permutations. This is a known issue (e.g. it is mentioned in the documentation for `GetBestRMS` in `rdkit`). Given that one claim made by the authors is that DMCG shows especially strong improvement for larger molecules (stated in section 4.2), I think that this potential limitation for large molecules should at the very least be mentioned, and ideally be thoughtfully discussed.
- **Potential errors in implementing loss function**: from the description in section A.1, it appears that the set of symmetries is defined only using information about the connectivity of a node's immediate neighbours. This is incorrect; it is easy to come up with counterexamples of 2 atoms in a molecule with identical immediate neighbours but for which there is no symmetry. If this implementation detail is indeed incorrect, it could potentially invalidate many of the results in the paper... If symmetries were identified with `rdkit` (e.g. the functions behind `GetBestRMS`) then I would have more confidence.
- **Some unclear experimental details**:
    - Which baselines did the authors run themselves, and for which baselines did the authors copy numbers from the original paper?
    - Were the number of parameters in the baseline methods scaled when moving from the small to large datasets? Since DMCG is scaled it seems fair that the number of parameters for all methods should be adjusted to e.g. minimize overfitting.
- **Space limitations**: in several places the authors mention "space limitations" as the reason for not including something in the main text of the paper. While I appreciate the honesty of the authors, I think that this should be avoided, especially in TMLR where submissions can be of any length. This was especially surprising since the submission main text is currently just over 11 pages, which is less than the 12 page maximum required for fast review. I think that the entirety of appendix A could (and probably should) be moved to the main text since these are important details of the paper.

---

> ### Author Response · Authors · 2022-08-22
> **Response to Reviewer PiFm [Part 1]**
>
> Thanks for your valuable review comments!
> > [Q1] (A. from requested changes) Clarification of whether the algorithm to determine molecular symmetries described in section a.1 is incorrect, or whether I have misunderstood it. (B. from weakness) it appears that the set of symmetries is defined only using information about the connectivity of a node's immediate neighbours. This is incorrect; it is easy to come up with counterexamples of 2 atoms in a molecule with identical immediate neighbours but for which there is no symmetry. If this implementation detail is indeed incorrect, it could potentially invalidate many of the results in the paper... If symmetries were identified
>
> Thanks for pointing this. Perhaps you misunderstood our method. The definition is built upon graph isomorphism, which is not equivalent to "only using information about the connectivity of a node's immediate neighbours". We prepare a supplementary material (SymmetrySearch.pdf) to help understand this definition. This supplementary file will also be public so that more readers can benefit from it.
>
> > [Q2] Potential poor scaling of loss not discussed:
>
> Thanks for your valuable suggestions! Yes, you are right, the problem indeed exists. When the number of permutation invariant mappings in a molecule is extremely large, enumerating all of them is not the best choice due to the exponentially increased computation cost. The priors about local structures should be considered and we leave it as the future work. (this discussion is added in "Conclusions and future work")
> For your comment " Given that one claim made by the authors is that DMCG shows especially strong improvement for larger molecules (stated in section 4.2),", we add a condition that "For the molecules in GEOM-QM9 and GEOM-Drugs} our method achieves more improvement on molecules with more heavy atoms."
>
> > [Q3] Remove "space limitation" stuff and move appendix A into the main text
>
> We remove all "space limitation". The reason we did not provide all contents in the main paper is that, we want to increase the readiness of the paper. If we put everything in the main paper, it will be too heavy for readers to capture the main idea. Thanks for your suggestion.
>
> Yes, we move Appendix A to the main content, Pleas kindly check the "Solver of Eqn.(1)" in Section 2.1.
>
> > [Q4] You use the phrase "expressiveness of our loss function": what does this mean? I don't think that losses "express" anything... I would like to see this re-phrased to be clearer.
>
> Sorry for the misleading. We revise it to "Based on the new loss function".
>
> > [Q5] The equations for sampling from a standard normal distribution on page 3 are only correct if $\Sigma$ is diagonal and if it is interpreted as standard deviation instead of variance (which is more common). This should be updated/clarified.
>
> Thanks for the information. We have updated this part in our draft.

---

> > ### Author Response · Authors · 2022-08-22
> > **Response to Reviewer PiFm [Part 2]**
> >
> > > [Q6] DMCG (and other similar methods) depend on having a large dataset of high-quality conformers, which is often unavailable in practice. It would be very good to comment on how the method's performance changes with dataset size (perhaps an additional experiment), or at the very least discuss this limitation explicitly in the paper.
> >
> > To investigate whether our method relies on a large dataset, we subsample training data of the large-scale GEOM-QM9 and GEOM-Drugs to 10\%, 25\%, 50\%, 75\%. The validation and test sets remain unchanged. The results are shown below:
> > | Dataset |                 QM9 |        |                        |        | &nbsp; &nbsp; &nbsp; &nbsp;&nbsp; &nbsp; &nbsp; &nbsp; |               Drugs |        |                        |        |
> > | ------- | ------------------: | :----- | ---------------------: | :----- | ------------------------------------------------------ | ------------------: | :----- | ---------------------: | :----- |
> > | Metric  | Cov($\\%$)$\uparrow$ |        | MAT (A)$\downarrow$ |        |                                                        | Cov($\\%$)$\uparrow$ |        | MAT(A)$\downarrow$ |        |
> > | Subset  |                Mean | Median |                   Mean | Median |                                                        |                Mean | Median |                   Mean | Median |
> > | ConfGF   |      89.21  | 95.12 | 0.2809 | 0.2837 || 70.92 |85.71 | 1.0940 | 1.0917 |
> > | GeoMol   | 91.05 | 95.55 |  0.2970 | 0.2993 || 69.74 | 83.56 | 1.1110 | 1.0864 |
> > | DMCG ($10\\%$)  |               91.48 | 97.45  |                 0.2748 | 0.2692 |                                                        |               94.47 | 100.00 |                 0.7934 | 0.7624 |
> > | DMCG  ($25\\%$) |               97.65 | 100.00 |                 0.1858 | 0.1676 |                                                        |               95.17 | 100.00 |                 0.7475 | 0.7092 |
> > | DMCG ($50\\%$)  |               98.23 | 100.00 |                 0.1606 | 0.1457 |                                                        |               96.38 | 100.00 |                 0.7057 | 0.6771 |
> > | DMCG ($75\\%$)  |               98.31 | 100.00 |                 0.1544 | 0.1384 |                                                        |               96.14 | 100.00 |                 0.6947 | 0.6562 |
> > | DMCG ($100\\%$) |               98.34 | 100.00 |                 0.1486 | 0.1340 |                                                        |               96.22 | 100.00 |                 0.6967 | 0.6552 |
> >
> >
> > We can see that:
> > - Generally, DMCG benefits from more training data.
> > - With $10\\%$ training data, our method is better than previous baselines.
> >
> >
> > > [Q7] You never explicitly define the acronym DMCG; it just appears in your github link and in the legend of Figure 4. You should really change this.
> >
> > Sorry for that. We explicitly define it in the introduction section, page 2.
> >
> >
> > >[Q8] Some unclear experimental details
> > - For the small scale GEOM-QM9 and small scale GEOM-Drugs, the results of RDKit, CVGAE, GraphDG, CGCF and ConfGF are copied from ConfGF paper. The results of ConfVAE, DGSM and GeoDiff are copied from the original papers. The rest results both in small scale and large scale datasets are reproduced by us.
> >  - In the initial submission, when moving from small to large dataset, all the baselines have been tuned with different model size except GeoMol. In the author response phase, we have tuned the model size for GeoMol. The results are: 1) the results on GEOM-QM9 are unchanged. That is, using larger models does not bring better performance. 2) the results on GEOM-Drugs are improved to (69.74/83.56/1.1110/1.0864), which are close to ConfGF's, while are still significantly worse than ours. We have updated these results in our draft.

---

> > > ### Comment · Reviewer_PiFm · 2022-08-25
> > > **Addressed my concerns**
> > >
> > > Thanks, I feel the the additional experiments/discussion has mostly addressed my concerns. I think the paper could still be presented in a better way however, which I will address in a general comment.

---

### Review · Reviewer_D4R5 · 2022-07-21

**Summary Of Contributions:**

In this paper, the Authors introduce a new method for predicting molecular conformations. The method uses an autoencoder type of a network that generates conformations by encoding molecular graphs and predicting all atom positions at the end of the network. The loss function contains two components. One is responsible for measuring the distance between the ground truth conformation and the predicted conformation (RMSD), and the other one is the Kullback-Leibler divergence applied to the latent vector $z$ that encodes 3D atom positions and allows the model to sample multiple conformations. To make the model invariant to rotations, translations, and the permutation of symmetric atoms, a composition of two transformations is applied to the predicted conformation, and the loss function is computed for the maximum agreement of conformational poses over these transformations. The results of the proposed model are demonstrated on both small- and large-scale GEOM-QM9 and GEOM-Drugs datasets. Additionally, the usability of the method for molecular property prediction and molecular docking (as initial conformation generation) is considered.

**Broader Impact Concerns:**

I do not see any ethical concerns that should be raised here.

**Requested Changes:**

- Please, explain how the hyperparameters shown in Section B.1 of the supplementary materials were selected.
- The Authors say that “on average, a molecule has 5.9 symmetric substructures” in the GEOM-Drugs dataset. How were these substructures counted? Is this the number of the axes of symmetry? Can this number be converted into a percentage that is easier to interpret?
- For the coverage metric, $\delta$ was selected to be 0.5 and 1.25 for the GEOM-QM9 and GEOM-Drugs datasets, respectively. Why were these numbers selected and why are they different for each dataset? Could a plot showing COV under different deltas be added to the supplementary materials?
- Why was only a selection of models used in some of the experiments? For example, ConfGF was used in the docking experiment, and not GeoMol or GeoDiff.
- In the ablations study, Figure 7 shows metrics calculated after each layer. Was the loss function in this experiment constructed to optimize the conformation of each layer, or only the final conformation was compared to the ground-truth positions?
- Would it not be more fair for the experiment in Section C.3 (Supplementary Material) to reduce the number of parameters in the proposed method instead of increasing the number of parameters for the other method? I believe the size of hidden dimensions was selected to be optimal by the authors of the other method (ConfGF).

**Strengths And Weaknesses:**

Strengths:
- The implementation of the method is made public.
- The related work section includes many recent models for conformation prediction, including a work on prediction of active conformation in a binding pocket.
- The proposed method is able to generate multiple conformations because of the latent vector $z$ that adds randomness to the model.
- The roto-translational invariance is achieved by the construction of the loss function that uses a transformation that aligns ground-truth and predicted conformations. The calculations are based on the quaternion method.
- The additional transformation that permutes symmetric atoms improves the results of the direct conformation prediction (significant contribution).
- The experimental section shows state-of-the-art results on the task of prediction molecular conformations.
- The ablation study justifies the architectural choices made.
- Additional experiments show that atom positions predicted by the method can be used for molecular prediction or as initial conformations for molecular docking.
- The paper is written in a clear way, with clear mathematical notation.
- The figures included in the paper help to grasp the idea of the method.
- The supplementary materials contain many more experiments and metrics.


Weaknesses:
- Neither confidence intervals nor standard deviations are provided for the results.
- The paper does not include a detailed discussion about computational cost (e.g. complexity of calculating transformations included in the loss function) though it argues that the method requires less iterations than other methods that can be found in the literature.
- There is no description of the hyperparameter tuning process. If the best set of hyperparameters was chosen based on the experimental testing results, then this might indicate a potential data leakage and unfair comparison with other methods.
- Some of the claims are not properly supported or ambiguous (see the requested changes below).

---

> ### Author Response · Authors · 2022-08-22
> **Response to Reviewer D4R5 [Part 1]**
>
> Thanks for your valuable review comments!
>
> > [Q1] Please, explain how the hyperparameters shown in Section B.1 of the supplementary materials were selected.
>
> The parameters are selected according to validation performance.
>
> > [Q2] The Authors say that "on average, a molecule has 5.9 symmetric substructures" in the GEOM-Drugs dataset. How were these substructures counted? Is this the number of the axes of symmetry? Can this number be converted into a percentage that is easier to interpret?
>
> The "5.9" means that the average number of the $\vert\mathcal{S}\vert$ for all molecules in GEOM-Drugs is 5.9. We revise the text as follows: According to our statistics on a subset of 40K molecules from GEOM Drugs, on average, each molecule has $5.9$ atom mappings which could result in the same conformation (more specifically, the average number of $\vert\mathcal{S}\vert$ in Eqn.(1) is $5.9$). The number is non-negligible for loss function design.
>
>
> > [Q3] For the coverage metric, $\delta$ was selected to be 0.5 and 1.25 for the GEOM-QM9 and GEOM-Drugs datasets, respectively. Why were these numbers selected and why are they different for each dataset? Could a plot showing COV under different deltas be added to the supplementary materials?
>
>
> The 0.5 and 1.25 are chosen following previous work like GeoDiff, ConfGF, DGSM, etc. Most of previous works use these two scores. Following your suggestions, we add the curves in Figure 9.
>
> > [Q4] Why was only a selection of models used in some of the experiments? For example, ConfGF was used in the docking experiment, and not GeoMol or GeoDiff.
>
> This is because we have limited resources, and it is difficult to run all baseline methods for each experiment. Now we add GeoMol and GeoDiff to the docking experiment. Our method is still the best. Please kindly refer to Figure 6.
>
> > [Q5] In the ablations study, Figure 7 shows metrics calculated after each layer. Was the loss function in this experiment constructed to optimize the conformation of each layer, or only the final conformation was compared to the ground-truth positions?
>
> It is "optimize the conformation of each layer". We give a more comprehensive description of the new loss function in Eqn.(8) of Section 4.1.
>
>
> > [Q6] Would it not be more fair for the experiment in Section C.3 (Supplementary Material) to reduce the number of parameters in the proposed method instead of increasing the number of parameters for the other method? I believe the size of hidden dimensions was selected to be optimal by the authors of the other method (ConfGF).
>
> | Dataset         |                 QM9 |        |                        |        | &nbsp; &nbsp; &nbsp; &nbsp;&nbsp; &nbsp; &nbsp; &nbsp; |               Drugs |        |                        |        |
> | --------------- | ------------------: | :----- | ---------------------: | :----- | ------------------------------------------------------ | ------------------: | :----- | ---------------------: | :----- |
> | Metric          | Cov($\\%$)$\uparrow$ |        | MAT(A)$\downarrow$ |        |                                                        | Cov($\\%$)$\uparrow$ |        | MAT(A)$\downarrow$ |        |
> | Methods         |                Mean | Median |                   Mean | Median |                                                        |                Mean | Median |                   Mean | Median |
> | ConfGF (0.81M)  |               88.49 | 94.13  |                 0.2673 | 0.2685 |                                                        |               62.15 | 70.93  |                 1.1629 | 1.1596 |
> | ConfGF (12.28M) |               86.86 | 93.49  |                 0.3377 | 0.3450 |                                                        |               55.36 | 58.20  |                 1.2186 | 1.2134 |
> | DMCG (13.29M)  |   96.23 | 99.26 | 0.2083 | 0.2014 || 96.52 | 100.00 | 0.7220| 0.7161 |
> | DMCG (0.98M)   |               94.28 | 98.20  |                 0.2399 | 0.2361 |                                                        |               89.40 | 97.06  |                 0.8653 | 0.8670 |
>
> Thanks for your suggestions.  We now reduce the number of parameters of our method to 0.98M. The results are still better than ConfGF.
>
> > [Q7] Neither confidence intervals nor standard deviations are provided for the results.
>
> We did not report the confidence intervals because previous work did not (GeoMol, GeoDiff). Following your suggestion, in Table 1, we add the standard deviation of our method.  The standard derivations of our method are significantly smaller than the gain compared to the previously best results. This shows the effectiveness and robustness of our method.

---

> > ### Author Response · Authors · 2022-08-22
> > **Response to Reviewer D4R5 [Part 2]**
> >
> > > [Q8] The paper does not include a detailed discussion about computational cost (e.g. complexity of calculating transformations included in the loss function) though it argues that the method requires less iterations than other methods that can be found in the literature.
> >
> > We use PyTorch profiler to analyze the training time of the following components: (1) model forward time, which denotes the time of calculating the hidden representations from the input layer to output layer; (2) transformation time, which denotes the time of calculating the optimal roto-translation operation $\rho^*$; (3) permutation time, which denotes the time of enumerating all possible permutations in $\mathcal{S}$ and find the optimal one $\sigma^* \in \mathcal{S}$; note that we can use \texttt{torch.no\_grad} to reduce time and memory;  (4) loss forward time, which is  the total of calculating the loss after obtaining $\rho^*$ and $\sigma^*$; (5) loss backward time, which denotes the time of gradient backpropagation.}
> >
> > The time is summarized in the following table. We can see that model forward and loss forward/backward takes about 71.4% of the total computation time. The transformation and permutation takes 20.4% and 8.2% of the total time. Note that there are $7$ transformation operations in the experiments (see Eqn.(8)). For the full training pipeline where data loading, model forwarding, loss forwarding, gradient backpropagation, metric calculation and CPU/GPU communications are all considered, DMCG takes 20% more time than that without roto-translation and permutation.
> >
> > |           Name |   CUDA time | Percentage  | # of Calls |
> > | -------------: |   ------------: |------------: | ---------: |
> > | Model Forward |     5.515s       | 52.8%   |    100 |
> > | Transformation |         2.136s |  20.4%|      700 |
> > | Permutation |     0.858s       |   8.2%|     100 |
> > | Loss foward |            0.052s |  0.5%|      100 |
> > | Loss backward |          1.886s |   18.1%|     100 |
> >
> >
> > The detailed discussion is left in Section 4.2.
> >
> > > [Q9] There is no description of the hyperparameter tuning process. If the best set of hyperparameters was chosen based on the experimental testing results, then this might indicate a potential data leakage and unfair comparison with other methods.
> >
> > All hyperparameters are selected according to validation performance.

---

> ### Comment · Reviewer_D4R5 · 2022-09-04
> **Thank you for your response**
>
> Thank you for your response. This addresses my concerns.

---

### Review · Reviewer_yvBx · 2022-08-07

**Summary Of Contributions:**

This paper proposes a method for generating 3D coordinates of atoms of a molecule. Their construction rests on a loss function that is invariant to symmetric molecule perturbations and rotations and translations of the molecular conformation. This loss function is implemented via enumeration of perturbations and an eigenvalue minimization problem using quaternions as presented in Karney 2007.

The  molecular generation architecture  consists  of  three  components:  (1)  an  encoder,  j2D,  that  takes  a  molecular graph as input and produces representations of all atoms (HV), covalent bonds (HE),  global graph-level features (U), and initial conformation (R); (2) another encoder, j3D, that takes  ground truth conformation and molecular graph as input and generates a normal distribution for
conditional inference; and (3) a decoder, jdec, that takes learned representations and samples  from the normal distribution to produce the final conformation. The encoders and decoder are  implemented using a stacked architecture, which iteratively updates HV, HE, U and R. The loss  formulation consists of a KL divergence term (to drive the marginalized posterior distribution of  the random variable towards the prior) and a reconstruction/matching term that enforces roto-translation and permutation invariance.

Numerical experiments are performed using small-scale and large-scale versions of GEOM-QM9  and GEOM-Drugs. Models are evaluated using mean coverage score (abbr. COV, higher is better)  and matching score (abbr. MAT, lower is better) based on RMSD between the predicted and  actual  conformations.  Additionally,  the  authors  compare  the  ensemble  averaged  quantum  properties  of  the  generated  conformations  and  ground  truth  conformations  (both  computed  using Psi4) via the mean absolute error. They also evaluate the docking performance (using 100 randomly sampled protein-ligand pairs) of the generated conformations.

**Broader Impact Concerns:**

I don't see any ethical implications that merit adding such a statement.

**Requested Changes:**

Requested changes mirror the weaknesses:
1. Discuss in more detail what prior work has been done in conformation generation including Hoogeboom et al.
2. More clearly motivate the advantage of direct 3D generation.
3. Motivate invariance vs equivariance and also add ablation studies that show the impact of ablating permutation and rototranslation invariance or relaxing to equivariance.
4. Discuss the details and implementation of the loss function in the main paper---this is a key part of this work. Perform complexity analysis of this, and what the gradients miss when they "skip" the optimization. Is it possible to use another neural network to compute the loss function?.
5. Compare stacked to recurrent architecture.
6. Add comparisons to the above mentioned methods.

Edits:
• Some of the models used for comparing recall-based COV and MAT
scores (Table 1) are missing from precision-based scores (Table 5).
• The authors mention that the code for DGSM is not available. However, COV and MAT
scores for DGSM are reported in Table 1. In fact, almost all values reported in Table 1 for
small-scale datasets are identical to Table 1 of the DGSM paper, “Predicting Molecular
Conformation via Dynamic Graph Score Matching” by Luo et al. However, the values
reported in Table 2 (MAE of property prediction) are very different from those reported
in Table 2 of the DGSM paper, especially for GraphDG. Is this difference due to the
random sampling of 30 molecules and 50 conformations?
• Although the COV score flattened out, the MAT score is still decreasing in Figure 7. Is
there room for further improvement by adding more blocks?
• In Figure 4, the proposed model is labelled as DMCG. This is inconsistent with other
figures and tables, where they refer to their model as “Ours”.
• In Table 5 and Table 8, the best result is not bolded.
• In Appendix C.2, Eqns. 12, 13 and 14, colored LaTeX is not needed.

**Strengths And Weaknesses:**

Strengths:
1. This method directly generates 3D coordinates, which can then immediately be used for docking, property prediction and virtual screening.
2. The method features a loss function that is designed for permutation and rotation/translation invariance using a combination of literature from quaternions and graph isomorphism,  utilized in this novel context. Moreover the loss function itself involves optimizations, 1) one which is finding a rotation minimizes the loss, is solved analytically and, 2) another which finds the minimizing permutation and solved by  enumeration. This idea of using optimizations to compute the loss function (but not backpropagating through them) is interesting but not highlighted in the manuscript.
3. The proposed method is effective for predicting conformations of large molecules,  especially those with heavier atoms. Decoding is much more efficient compared to  sequential methods of generation, and the conformations generated are free from autocorrelation.

Weaknesses:

1. The introduction tries to motivate direct molecular generation by listing methods which generate intermediate quantities such as  interatomic distances, gradients wrt interatomic distances, torsion angles first. However, it is not really clear what is wrong with the latter approach. Indeed many of these quantities, such as relative distances and angles are invariant to rotation/translation already.
2. The paper assumes that permutation and roto-translation invariance is a requirement for training a neural network to generate molecules, but this is not actually clear. What if a more relaxed condition like equivariance was sought [see Hoogeboom et al. ICML 2022 "Equivariant Diffusion for Molecule Generation" or Villar et al. NeurIPS 2021 "Scalars are universal: Equivariant machine learning, structured like classical physics"].  In fact, even with equivariance, it is not clear if it creates a more desirable solution space.
3. The paper basically ignores its main contribution and does not discuss it at all the implications of their proposed loss function. Is it effective to compute? What is the effect of not backpropagating through the optimizations? Graph isomorphism is NP-hard, so how does this effect the solutions?  Are these choices ablated. The loss function and its solution are not even explained in the main paper.
4. The authors propose a stacked or iterative architecture. Why is this needed? Why not have this be recursive?
5. Lack of comparisons to methods that directly generate 3D conformations including the above mentioned paper from ICML 2022 as well as Clevees & Jain https://link.springer.com/article/10.1007/s10822-017-0015-8.
6. Comparison to non-ML method OMEGA is not performed, OMEGA is SOTA and outperforms RDKit

---

> ### Author Response · Authors · 2022-08-22
> **Response to Reviewer yvBx [Part 1]**
>
> Thanks for your valuable review comments!
> > [Q1] Discuss in more detail what prior work has been done in conformation generation including Hoogeboom et al.
>
> Thanks for the three references and we cite them. Specifically,
>
> 1. Our method models the roto-translation and permutation invariance through the loss function, while previous works model the molecules using equivariant networks (Hoogeboom et al., 2022; Xu et al., 2022). More specifically, these works used a diffusion model for conformation generation. Rotational invariance of the conformation distribution is implemented using an invariant latent prior and an equivariant model structure (reverse diffusion process) to map from the latent space to the conformation space. This effectively makes an invariant loss in the **latent space**. Hoogeboom et al. (2022) also generate the composition of a molecule, by leveraging continuous representation of ordinal/categorical variables.
> In comparison, our method is a different way to implement invariance. We employ an invariant loss function directly on the **conformation space** to measure the essential difference between the predicted conformation and the ground-truth. When used in the likelihood distribution, it naturally defines a permutational and roto-translational invariant conformation distribution, using **any** model structure. The relaxation of equivariance allows using potentially more powerful models. We add the discussion in Section 3.
>
> 2. The main conclusion of Villar et al. is that "...it is simple to parameterize universally approximating polynomial functions that are equivariant under these symmetries" (words from the original paper). This is an insightful paper and will be helpful for incorporating various equivariant/invariant properties into neural networks. We will implement it to improve our model backbone. However, another contribution of our paper is to use a roto-translational and permutational invariance loss function, which is complementary to this work. This is discussed in the "Conclusions and future work" section.
> 3. (Clevees & Jain 2017) proposed another forcefield driven method for conformation generation and we cite them in the introduction part.
> We try to apply for a license but it requires that "The free academic version is unsupported and is available for users at qualifying academic or non-profit research institutions within the United States and for qualifying users within the following countries, all of which are active members within the Organization for Economic Co-operation and Development: [several country names]" Unfortunately, the authors' country is not in the list.
>
> > [Q2] More clearly motivate the advantage of direct 3D generation.
>
> Previous methods (e.g., ConfGF, DSGM, GeoMol) definitely improve molecular conformation generation, however, relying on the intermediate values brings difficulties for end-to-end training and could introduce accumulated error. More importantly, these intermediate values might violate the constraints like triangle inequality of distances, the degree of freedom of the distance matrix and rank of squared distance matrix (see Appendix C of the paper for the details). Directly generating the coordinates without those intermediate values is a more straightforward strategy but is not fully explored.
>
> The revision is posted at the 2nd paragraph of "introduction".
>
> > [Q3] Motivate invariance vs equivariance and also add ablation studies that show the impact of ablating permutation and rototranslation invariance or relaxing to equivariance.
>
> Please refer to the first item of our reply to your [Q1].
>
> The ablation study about permutation invariance is in Section 4.5 (first one). Without roto-translational loss function, the training cannot converge.

---

> > ### Author Response · Authors · 2022-08-22
> > **Response to Reviewer yvBx [Part 2]**
> >
> > > [Q4] (1) Discuss the details and implementation of the loss function in the main paper---this is a key part of this work. (2) Perform complexity analysis of this, and (3) what the gradients miss when they "skip" the optimization. (4) Is it possible to use another neural network to compute the loss function?.
> >
> > To (Q4-1) We move the solver of the loss function from the Appendix to Section 2.1. Please kindly have a check.
> >
> > To (Q4-2): We use `PyTorch profiler` to analyze the training time of the following components: (1) model forward time, which denotes the time of calculating the hidden representations from the input layer to output layer; (2) transformation time, which denotes the time of calculating the optimal roto-translation operation $\rho^*$; (3) permutation time, which denotes the time of enumerating all possible permutations in $\mathcal{S}$ and find the optimal one $\sigma^* \in \mathcal{S}$; note that we can use `torch.no_grad` to reduce time and memory;  (4) loss forward time, which is  the total of calculating the loss after obtaining $\rho^*$ and $\sigma^*$; (5) loss backward time, which denotes the time of gradient backpropagation.
> >
> > The time is summarized in the following table. We can see that model forward and loss forward/backward takes about 71.4% of the total computation time. The transformation and permutation takes 20.4% and 8.2% of the total time. Note that there are 7 transformation operations in the experiments (see Eqn.(8)). For the full training pipeline where data loading, model forwarding, loss forwarding, gradient backpropagation, metric calculation and CPU/GPU communications are all considered, DMCG takes 20% more time than that without roto-translation and permutation.
> >
> >
> > |           Name |   CUDA time | Percentage  | # of Calls |
> > | -------------: |   ------------: |------------: | ---------: |
> > | Model Forward |     5.515s       | 52.8%   |    100 |
> > | Transformation |         2.136s |  20.4%|      700 |
> > | Permutation |     0.858s       |   8.2%|     100 |
> > | Loss foward |            0.052s |  0.5%|      100 |
> > | Loss backward |          1.886s |   18.1%|     100 |
> >
> > The detailed discussion is left in Section 4.2.
> >
> > To (Q4-3): We "stop backprop" the gradient from $\rho$, because the official document of "torch.linalg.eig" has a warning: Gradients computed using the eigenvectors tensor will only be finite when $A$ has distinct eigenvalues. Furthermore, if the distance between any two eigenvalues is close to zero, the gradient will be numerically unstable, as it depends on the eigenvalues $\lambda_i$ through the computation of $\frac{1}{\min_{i \neq j} \lambda_i - \lambda_j}$.
> >
> > We also compare the results with or without gradients on $\rho$.
> >
> > | Dataset    |                 QM9 |        |                        |        | &nbsp; &nbsp; &nbsp; &nbsp;&nbsp; &nbsp; &nbsp; &nbsp; |               Drugs |        |                        |        |
> > | ---------- | ------------------: | :----- | ---------------------: | :----- | ------------------------------------------------------ | ------------------: | :----- | ---------------------: | :----- |
> > | Metric     | Cov($\\%$)$\uparrow$ |        | MAT(A)$\downarrow$ |        |                                                        | Cov($\\%$)$\uparrow$ |        | MAT(A)$\downarrow$ |        |
> > | Name       |                Mean | Median |                   Mean | Median |                                                        |                Mean | Median |                   Mean | Median |
> > | With Grad |               96.14 | 99.43  |                 0.2090 | 0.2062 |                                                        |               96.08 | 100.00 |                 0.7345 | 0.7247 |
> > | Stop Grad  |   96.23 | 99.26 | 0.2083 | 0.2014 || 96.52 | 100.00 | 0.7220| 0.7161 |
> >
> > We can see that the performance of enabling the gradients slightly drops.
> > Therefore, overall, we recommend disabling	 the gradients over $\rho$.
> > The discussion is in Section B.3 of Appendix.
> >
> > To (Q4-4):  We do not think it is necessary because the loss function has analytical solutions.

---

> > > ### Author Response · Authors · 2022-08-22
> > > **Response to Reviewer yvBx [Part 3]**
> > >
> > > > [Q5] Compare stacked to recurrent/recursive architecture.
> > > Following your suggestion, we also implement a recursive version, where in the decoder side, the parameters of different blocks are shared.
> > >
> > > | Dataset   |                 QM9 |        |                        |        | &nbsp; &nbsp; &nbsp; &nbsp;&nbsp; &nbsp; &nbsp; &nbsp; |               Drugs |        |                        |        |
> > > | --------- | ------------------: | :----- | ---------------------: | :----- | ------------------------------------------------------ | ------------------: | :----- | ---------------------: | :----- |
> > > | Metric    | Cov($\\%$)$\uparrow$ |        | MAT(A)$\downarrow$ |        |                                                        | Cov($\\%$)$\uparrow$ |        | MAT(A)$\downarrow$ |        |
> > > | Name      |                Mean | Median |                   Mean | Median |                                                        |                Mean | Median |                   Mean | Median |
> > > | DMCG| 96.23 | 99.26 | 0.2083 | 0.2014 || 96.52 | 100.00 | 0.7220| 0.7161 |
> > > | DMCG with recursive decoder|               94.03 | 98.00  |                 0.2726 | 0.2771 |                                                        |               95.20 | 100.00 |                 0.7883 | 0.7862 |
> > >
> > > We can see that the performances drop.
> > > We add the discussion in Section B.9 of Appendix.
> > >
> > > > [Q6] Add comparisons to the above mentioned methods.
> > >
> > > Please refer to the reply to [Q1].
> > >
> > > > [Q7] The introduction tries to motivate direct molecular generation by listing methods which generate intermediate quantities such as interatomic distances, gradients wrt interatomic distances, torsion angles first. However, it is not really clear what is wrong with the latter approach. Indeed many of these quantities, such as relative distances and angles are invariant to rotation/translation already.
> > >
> > > Please refer to the reply to [Q2].
> > >
> > > > [Q8] The paper assumes that permutation and roto-translation invariance is a requirement for training a neural network to generate molecules, but this is not actually clear. What if a more relaxed condition like equivariance was sought [see Hoogeboom et al. ICML 2022 "Equivariant Diffusion for Molecule Generation" or Villar et al. NeurIPS 2021 "Scalars are universal: Equivariant machine learning, structured like classical physics"]. In fact, even with equivariance, it is not clear if it creates a more desirable solution space.
> > >
> > > For the comparison with the two work you listed, please kindly refer to our reply to [Q1].  Considering that Hoogeboom et al. ICML 2022 does not directly work on 2D molecules to 3D conformation generation, we compare with a similar work GeoDiff (Xu et al, 2022), which also used equivariant network and diffusion model for conformation generation. Our method is better than (Xu et al 2022).
> > >
> > >
> > > > [Q9] The paper basically ignores its main contribution and does not discuss it at all the implications of their proposed loss function. (1) Is it effective to compute? (2) What is the effect of not backpropagating through the optimizations? (3) Graph isomorphism is NP-hard, so how does this effect the solutions?  (4) Are these choices ablated. The loss function and its solution are not even explained in the main paper.
> > >
> > > To (Q9-1): Please refer to the reply to (Q4-2).
> > >
> > > To (Q9-2): Please refer to the reply to (Q4-3).
> > >
> > > To (Q9-3): Although the general graph isomorphism problem is NP-hard, the size of drug-like molecules is largely limited, otherwise the molecule's druggability is limited (one can refer to Lipinski's rule of five). Therefore, our method does not need scalability to a large scale. In our experiments, the graph isomorphism computation takes $4.9$ seconds to process $10k$ molecules in GEOM-QM9, and $6.6$ seconds to process $10k$ molecules in GEOM-Drugs. This is negligible compared to the training time, and we only need to process the data for one time.
> > >
> > > To (Q9-4): We now move the detailed algorithm to Section 2.1. The ablation of loss function is in Section 4.5.
> > >
> > >
> > > > [Q10] The authors propose a stacked or iterative architecture. Why is this needed? Why not have this be recursive?
> > >
> > > Iteratively polishing the output has been verified to be useful in NLP (Xia et al., 2017), CV (Chen & Koltun, 2017) and AlphaFold 2 (Jumper et al., 2021). Therefore, we choose to use it in our work.
> > >
> > > For experiment comparison, please refer to our reply to [Q5].

---

> > > > ### Author Response · Authors · 2022-08-22
> > > > **Response to Reviewer yvBx [Part 4]**
> > > >
> > > > > [Q11] Lack of comparisons to methods that directly generate 3D conformations including the above mentioned paper from ICML 2022 as well as Clevees & Jain https://link.springer.com/article/10.1007/s10822-017-0015-8.
> > > >
> > > > For the ICML22 paper, please kindly refer to the reply to [Q1].
> > > > For (Clevees & Jain 2017), we cannot get the license (refer to our reply to [Q1] (3rd item)). Actually, none of GeoMol, GeoDiff, ConfGF compare with it.
> > > >
> > > >
> > > > > [Q12] Comparison to non-ML method OMEGA is not performed, OMEGA is SOTA and outperforms RDKit
> > > >
> > > > We have applied for the academic  license for OMEGA,  and it will takes approximately 2 to 4 weeks to determine eligibility. We will compare with OMEGA after we get the academic license.
> > > >
> > > > > [Q13]  Some of the models used for comparing recall-based COV and MAT scores (Table 1) are missing from precision-based scores (Table 5)
> > > >
> > > > Because the precision-based is a new metric, and many prior works did not report this score. We collect the results from the GeoDiff paper and summarize them in Table 5.
> > > >
> > > > > [Q14] The authors mention that the code for DGSM is not available. However, COV and MAT scores for DGSM are reported in Table 1. In fact, almost all values reported in Table 1 for small-scale datasets are identical to Table 1 of the DGSM paper, "Predicting Molecular Conformation via Dynamic Graph Score Matching" by Luo et al. However, the values reported in Table 2 (MAE of property prediction) are very different from those reported in Table 2 of the DGSM paper, especially for GraphDG. Is this difference due to the random sampling of 30 molecules and 50 conformations?
> > > >
> > > > For Table 1, the authors did not release the code. However, they use the same data split as the previous work. Therefore, we can cite the numbers from their papers.
> > > > For Table 2, the differences of properties are mainly due to the random sampling of 30 molecules. We have also asked the SMILES of 30 molecules used in ConfGF (ConfGF and DGSM have the same authors) via email when we conducted this experiment, and have not received any response.
> > > >
> > > > > [Q15] Although the COV score flattened out, the MAT score is still decreasing in Figure 7. Is there room for further improvement by adding more blocks.
> > > >
> > > > We increase the number of blocks from $6$ to $12$. The results are shown below. For GEOM-QM9, we do not observe performance improvement by increasing the number of blocks. For GEOM-Drugs, increasing the number of blocks further improves the performance. Our conjecture is that, the molecules in GEOM-Drugs are more complex than those in GEOM-QM9, which benefit more from larger models. The results are updated in Table 13 of Appendix.
> > > >
> > > > | Dataset  |                 QM9 |        |                        |        | &nbsp; &nbsp; &nbsp; &nbsp;&nbsp; &nbsp; &nbsp; &nbsp; |               Drugs |        |                        |        |
> > > > | -------- | ------------------: | :----- | ---------------------: | :----- | ------------------------------------------------------ | ------------------: | :----- | ---------------------: | :----- |
> > > > | Metric   | Cov($\\%$)$\uparrow$ |        | MAT(A)$\downarrow$ |        |                                                        | Cov($\\%$)$\uparrow$ |        | MAT(A)$\downarrow$ |        |
> > > > | # Layers |                Mean | Median |                   Mean | Median |                                                        |                Mean | Median |                   Mean | Median |
> > > > 6 | 96.23 | 99.26 | 0.2083 | 0.2014 || 96.52 | 100.00 | 0.7220| 0.7161 |
> > > > | 8        |               95.55 | 98.91  |                 0.2217 | 0.2190 |                                                        |               96.77 | 100.00 |                 0.7122 | 0.7093 |
> > > > | 10       |               95.71 | 99.52  |                 0.2215 | 0.2212 |                                                        |               97.29 | 100.00 |                 0.7089 | 0.7079 |
> > > > | 12       |               94.65 | 99.11  |                 0.2280 | 0.2284 |                                                        |               97.11 | 100.00 |                 0.7092 | 0.6996 |

---

### Decision · Action_Editors · 2022-10-05

**Recommendation:** Accept as is

**Comment:**

The paper proposes to generate molecular conformation by directly predicting atom positions. Core to the method is a principled loss function that is invariant with respect to rotation/translations as well as permutations of atom indices that do not change the molecule. The invariance to index permutation is achieved by computing the loss over different precomputed permutations.

The core idea is simple and the method achieves strong empirical performance according to a thorough evaluation. Among the compared approaches, the proposed method is the only one that consistently improves upon Rdkit. The high accuracy of the predicted conformation is shown to translate into downstream improvements when used as input to the calculation of molecular properties or the binding energy to proteins.

After a thorough discussion, reviewers unanimously recommended accepting the paper. The principled construction of the loss function and strong empirical performance was seen as key strength. Limited comparison to previous work and lack of clarity about picking hyperparameters were some of the discussed weaknesses.

Authors have released code and instructions on how to reproduce the results. During the review process, it was discovered and addressed that instructions are not fully complete. Overall, please make sure it is straightforward to rerun key experiments from the paper.

All in all, it is my pleasure to recommend acceptance of the paper. Thank you for submitting to TMLR, and please make sure to address all comments raised by reviewers. In particular, remember to include a more detailed discussion of how hyperparameters were chosen, and please add an ablation study that helps understand how the performance depends on the exact choice of hyperparameters.

**Audience:**

The paper makes a contribution to predicting the conformation of a molecule, which is a core property of a molecule that influences many of its properties. As such, it would be clearly interesting to the part of the TMLR audience interested in applications of machine learning to chemistry and biology (e.g. in the context of drug discovery).

**Claims And Evidence:**

The core idea is clear and simple and the method achieves strong empirical performance according to a thorough evaluation.

---

> ### Author Response · Authors · 2022-10-23
> **Camera-Ready Version**
>
> Dear AE and reviewers,
>
> Thank you for the time and effort you put into our work, and your suggestions have really improved the quality of our work. Moreover, we have made several updates in our camera-ready version.
>
> - We include the details about how to choose hyper-parameters in Appendix A.2.
> - We add an ablation study about the model size in  Appendix B.4.
>
> Best regards,
>
> The authors.